# Great GATsBi: Hybrid, Multimodal, Trajectory Forecasting for Bicycles using Anticipation Mechanism

## Abstract

Accurate prediction of road user movement is increasingly required by many applications ranging from advanced driver assistance systems to autonomous driving, and especially crucial for road safety. Even though most traffic accident fatalities account to bicycles, they have received little attention, as previous work focused mainly on pedestrians and motorized vehicles. In this work, we present the *Great GATsBi*, a domain-knowledge-based, hybrid, multimodal trajectory prediction framework for bicycles. The model incorporates both physics-based modeling (inspired by motorized vehicles) and social-based modeling (inspired by pedestrian movements) to explicitly account for the dual nature of bicycle movement. The social interactions are modeled with a graph attention network, and include decayed historical, but also anticipated, future trajectory data of a bicycles neighborhood, following recent insights from psychological and social studies. The results indicate that the proposed ensemble of physics models – performing well in the short-term predictions – and social models – performing well in the long-term predictions – exceeds state-of-the-art performance. We also conducted a controlled mass-cycling experiment to demonstrate the framework's performance when forecasting bicycle trajectories and modeling social interactions with road users.

## 1 Introduction

Accurate prediction of road user movement is crucial for various applications, including urban planning, infrastructure design, and intelligent transportation systems. Effective tracking and surveillance of road users can provide timely information, enabling smart signalized intersections (Yu et al., 2023). Moreover, precise forecasts of neighboring road users enhance safety systems in advanced driver assistance technologies, helping prevent collisions and accidents (Kosaraju et al., 2019). Autonomous vehicles, in particular, frequently interact with neighboring road users and require accurate perceptions of both current and future positions to navigate safely (Madjid et al., 2025).

Trends in urbanization, municipal sustainability efforts, increasing issues with congestion and emissions, and technological innovations drive the increasing use of active modes, like e-bikes and bicycles. Each year, road traffic accidents result in approximately 1.24 million fatalities, with vulnerable road users (VRUs) and especially cyclists comprising the largest proportion of these deaths (Toroyan et al., 2013; Gao et al., 2021; Zernetsch et al., 2016; Bessi, 2023). Despite the increasing trend in severe traffic accidents involving bicycles and VRUs (de Guerre et al., 2020; Juhra et al., 2012), their trajectory forecasting remains heavily underrepresented in the literature, that primarily focuses on pedestrians and motorized vehicles (Huang et al., 2022; Ding & Zhao, 2023; Schuetz & Flohr, 2023; Madjid et al., 2025).

Bicycle trajectory prediction is particularly challenging due to their hybrid behavioral characteristics. Bicycles and motorized vehicles alike, are subject to non-holonomic kinematic constraints and can reach relatively high speeds. Similarly, maneuver-related planning patterns such as following, lane changing and overtaking can be observed (Gao et al., 2021). However, unlike cars, bicycles are often less constrained by lane boundaries and can exhibit more flexible and sudden-change behaviors (in direction), similar to pedestrians (Arlauskas, 2025; Li et al., 2023; Brunner et al., 2024).

This combination of vehicle-like dynamics and pedestrian-like maneuverability complicates accurate trajectory prediction, especially in mixed-traffic environments (Li et al., 2023).

Previous works on bicycle trajectory forecasting primarily modeled bicycle behavior similar to pedestrians. By focusing on intent (Pool et al., 2019; Gao et al., 2021), sudden change behavior (Li et al., 2023) and multiple interactions with the neighborhood (Huang et al., 2021; Li et al., 2023) when modeling social context features, they neglect physical, kinematics-related properties of bicycles. Moreover, previous efforts are characterized by data-driven approaches only, overlooking the potential of incorporating domain-knowledge with learning, and impeding interpretability (Borghesi et al., 2020; Dash et al., 2022).

To address the challenges of bicycle trajectory forecasting and related limitations, we propose the great *GATsBi*, a hybrid, domain knowledge-based framework, that leverages both social and physical modeling, for multimodal trajectory forecasting of bicycles. First, we model the social context as a graph structure using graph attention networks (GATs) and explicitly incorporate anticipation and perception decay following recent insights from behavioral modeling (Hastie, 2022; Tarder-Stoll et al., 2024; Kinsky et al., 2020; Tanke et al., 2023; Ruan et al., 2024). Second, we combine kinematic and extended Kalman-filtering forecasting models to a deep ensemble embedding for the physics context. Third, we follow a hybrid approach, that combines these two, domain-knowledge-based contextual features, to gain improvements in forecasting and enabling insights for interpretability. Doing so, we account for the dual nature of bicycles, that share behavioral aspects with pedestrians and motorized vehicles.

We conducted a real-world, closed-loop, mass cycling experiment that enabled the observation of various interactions and bicycle maneuvers in varying traffic density contexts without the interference of road-contextual properties. A comparative benchmark on the trajectory dataset supports the contributions of the proposed method when compared with common forecasting methods for pedestrians and vehicles. The trajectory dataset, and implementation will be made publicly accessible in an open-source repository upon publication.

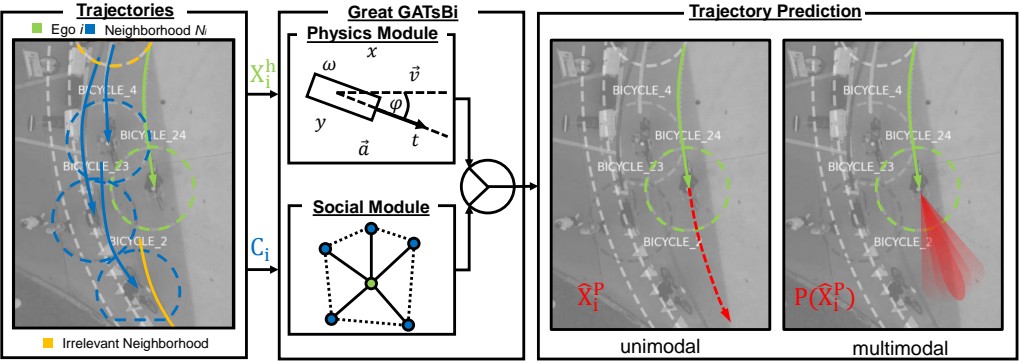

Figure 1: **Problem Statement: Trajectory Prediction.** This work is concerned with forecasting the trajectories of moving road users given their neighbors and their own (ego) historical trajectory information.

## 2 RELATED WORK

Trajectory forecasting typically involves contextual factors related to physics, road conditions, and social interactions. Prediction models can be categorized into physics-based methods, such a Kalman filtering approaches, and machine learning-based methods, which are better suited for capturing road and social context. These models can predict various types of outcomes, including a single trajectory (unimodal), a probabilistic distribution of possible future trajectories (multimodal), and driver intentions (maneuvers) (Huang et al., 2022). Recent works especially leveraged the potential of machine and deep learning (ML) based approaches (Ding & Zhao, 2023).

**Bicylce Trajectory Forecasting.** Contrary to previous work that relied on kinematic modeling approaches, Zernetsch et al. (2016) combined a simple physics model with a multi-layer perceptron

(MLP) to achieve better predictions. Pool et al. (2017) showed that combining road topology features with the bicycle's historical trajectory further enhances forecasts. Pool et al. (2019); Gao et al. (2021) highlighted the importance of learning the latent intention of the cyclist, which allows the model to leverage behavioral cues in trajectory prediction. Li et al. (2023) expanded on this by differentiating between intentions and latent sudden change behavior to better account for behavioral heterogeneity. Huang et al. (2021) demonstrated that incorporating neighborhood information to model multiple interactions between the cyclist and other road users can yield further improvements. Besides, a growing branch of work explored fast-inference, learning-based sensor fusion of onboard sensors for equipped, smart bicycles (Koornstra, 2023; Arlauskas, 2025; Sass et al., 2023; Bessi, 2023). Despite these advances, the former works are limited in three aspects. First, they neglected the multimodal nature of bicycle behavior which was shown to significantly improve forecasting accuracy in a pedestrian context (Kosaraju et al., 2019). Second, they focused on social context and overlooked the importance of systematic combination of physics-knowledge-based models. Third, their neighborhood was modeled over-simplistically in the form of focal attention mechanisms. To address these gaps, we propose a multimodal forecasting model based on Gaussian Mixture Models (GMM), explicitly combining physics-knowledge and social-knowledge to improve forecasting accuracy, and model the bicycle's social context as a graph using graph attention networks (GATs).

**Human Motion Modeling.** Traditionally, human motion modeling was often characterized by analytical modeling approaches. Notably, *Helbing's Social Force Model* (Helbing & Molnar, 1995) explicitly takes surrounding individuals and human-human interactions into account when modeling human motion. Modern approaches yield significantly improved forecasts leveraging ML approaches, such as *SocialLSTM* (Alahi et al., 2016). *Social-BiGAT* (Kosaraju et al., 2019) highlighted the importance of multimodality in the context of motion modeling, and explicitly reflected the social context in a graph neural network (GNN). In our model, we draw upon these advancements from pedestrian modeling and transfer them into the context of bicycles.

**Anticipation-Based Behavior Modeling.** Recent brain studies indicate human perception systems anticipate the actions of others. Consequently, studies from social sciences (Ng et al., 2022) predict human motions based on the surrounding actors' behaviors and interactional communication in dyadic conversations. Tanke et al. (2023) develops a *Social Diffusion Model* for motion forecasting, anticipating social signals based on contextual human poses. Ruan et al. (2024) presents a cooperative approach, where autonomous cars share local information for enhancing prediction accuracy, indicating the importance of shared perception in trajectory forecasting. Moreover, findings indicate that hippocampus-related memory systems are subject to decay (Hastie, 2022; Tarder-Stoll et al., 2024; Kinsky et al., 2020). Due to these findings and demonstrated advancements of behavioral anticipation in different domains, we explicitly include anticipation and perception decay to enhance the social context of trajectory forecasting beyond mere graph features, and thus to better extract social signals from the neighborhood.

**Knowledge-Based Machine Learning.** A growing branch of literature is concerned with incorporating domain knowledge into data-driven ML approaches, yielding better forecasts as ensemble and enhancing interpretability (Borghesi et al., 2020; Dash et al., 2022). In the context of trajectory prediction, social domain knowledge has been shown to yield improvements for pedestrians (Korbmacher & Tordeux, 2022; Huang et al., 2022), and physics domain knowledge has been shown to yield improvements for motorized road vehicles (Liao et al., 2024; Geng et al., 2023), and drones (Shukla et al., 2024; Bianchi et al., 2024) recently. Based on these insights, we aim to follow a hybrid approach and to combine social and physics domain-knowledge from pedestrians and vehicles to enhance bicycle trajectory forecasts.

## 3 METHODS: THE GREAT GATSBI

### 3.1 PROBLEM DEFINITION

The problem of trajectory forecasting manifests in the prediction $\mathcal{F}$ of entity $i$'s (hereafter called the *ego*) future movement $X_i^p$ through space, given its past trajectory $X_i^h$ and contextual information $\mathcal{C}_i$: $\hat{X}_i^p = \mathcal{F}\{X_i^h, \mathcal{C}_i\}$. Following previous works on pedestrians, bicycles, and motorized vehicles, we consider a two dimensional space with a historical observation horizon $t_{obs}$ and a prediction horizon $t_{pred}$, resulting in a past trajectory $X_i^h = \{(x_i^t, y_i^t) \in \mathbb{R}^2 | 0 \leq t \leq t_{obs}\}$ and a future trajectory

$X_i^p = \{(x_i^t, y_i^t) \in \mathbb{R}^2 | t_{obs} \leq t \leq t_{obs} + t_{pred}\}$. In this work, we consider social contextual information $C_i = C_i^S$ only, excluding road-contextual information $C_i^R$. The social contextual information consists of the past trajectory $X_j^h$ of the ego's neighborhood $\mathcal{C}_i^S = \bigcup_{j \in \mathcal{N}_i} X_j^h$ (neighboring road users of ego bicycle $i$). In the unimodal trajectory forecasting context, the prediction represents one trajectory $\mathcal{F} \in \mathbb{R}^2$, while in this work we consider the multimodal trajectory forecast, which is a probabilistic distribution across possible future trajectories $\mathcal{F} \in P(\mathbb{R}^2)$.

Common evaluation measures in the domain of trajectory prediction, that we will use in this work, include the average displacement error $ADE = \frac{1}{t_{pred}} \sum_t \|\hat{X}_i^p - X_i^p\|_2$, and the final displacement error $FDE = \|\hat{X}_{i,t_f}^p - X_{i,t_f}^p\|_2$ at time $t_f = t_{obs} + t_{pred}$.

## 3.2 MODEL ARCHITECTURE

The proposed model comprises four main modules, as shown in Figure 2. A *physics module* utilizes domain knowledge of the vehicle-like bicycle dynamics to predict the possible physics-based future ego trajectory and encodes it into the latent space: $Y_i^{\text{Phy}}$. The *social module* further uses the contextual information $C_i$ to encode the possible the ego's future trajectory into the same latent space: $Y_i^{\text{social}}$. The latent predictions, $Y_i^{\text{Phy}}$ and $Y_i^{\text{Soc}}$, are then amalgamated through the *fusion module* and decoded to $Z_i$. The *output module* finally transforms the result either in the unimodal prediction $\hat{X}_i^p$, or into the multimodal prediction $P(\hat{X}_i^p)$ in the form of Gaussian mixture models: $P(\hat{X}_i^p) = \{\mu_x, \mu_y, \sigma_x, \sigma_y, \rho, \pi\}$.

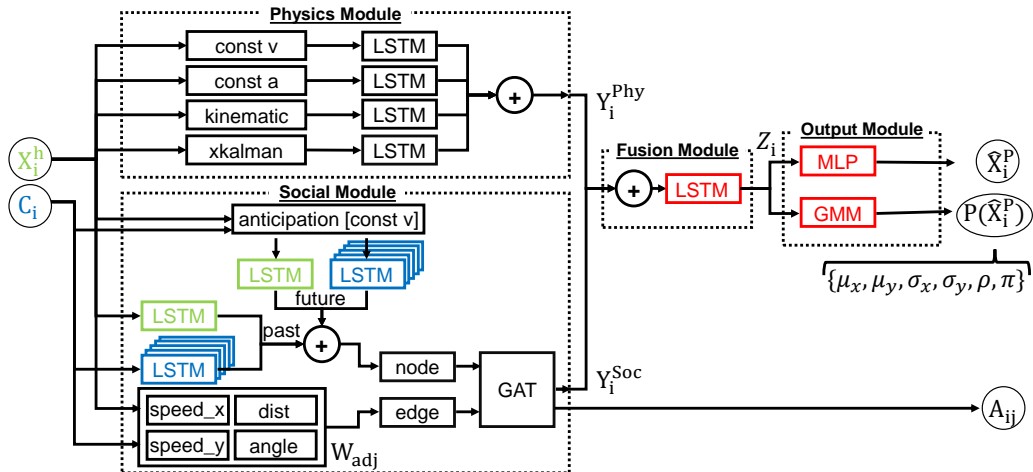

Figure 2: **Architecture for proposed Great GATsBi model.** The model consists of four modules. The physics module predicts the trajectory using common physical forecasting methods. The social module predicts the trajectory using a social graph structure, where the node features are a combination of past and anticipated future trajectories, and where the edge features are geometric adjacency features. The fusion module combines physics and social prediction. The output module transforms these forecasts back into the time domain, for unimodal or multimodal trajectory forecasting.

## 3.3 PHYSICS CONTEXT EMBEDDING

In order to exploit the vehicle-like dynamics of the bicycle and existing domain knowledge, we combine the knowledge of four common physics-based models. *const_v* assumes a constant velocity point kinematic model for the bicycle, where the velocity is estimated from the last two frames of the historical trajectory. *const_a*, assumes a constant acceleration point kinematic model. *kinematic* assumes simple bicycle kinematics, characterized by position $(x, y)$, direction $(\varphi)$, speed $(v)$:

$$\dot{x} = v \cos\varphi, \qquad \dot{y} = v \sin\varphi, \qquad \dot{\varphi} = \frac{v}{L_B} \tan\delta \qquad (1)$$

where $\delta$ is the steering angle and $L_B$ is the length of the bicycle. *xkalman* is an extended Kalman filtered (Kalman, 1960) prediction model based on the aforementioned kinematic model. All four physics-based forecasts for the ego trajectory are then encoded into latent space using separate long-short-term memory models (LSTMs). The encoded predictions are concatenated into the physical embeddings $Y_i^{\text{Phy}}$.

### 3.4 Social Context Embedding

Social interactions are modeled in the *social module*, following three assumptions based on recent findings in psychology and social sciences. First, human motions are affected by their neighborhood's social static context (Huang et al., 2021; Li et al., 2023; Alahi et al., 2016; Kosaraju et al., 2019). Second, humans anticipate the motion of their neighbors (Ng et al., 2022; Tanke et al., 2023; Ruan et al., 2024). Third, humans capabilities to memorize past and to anticipate future trajectories are limited for longer time horizons (Hastie, 2022; Tarder-Stoll et al., 2024; Kinsky et al., 2020). To this end, we incorporate the following mechanisms when extracting contextual embeddings.

**Perception decay.** We give higher temporal "attention" to the sections of the past and anticipated trajectories closest to the current timestep. Therefore, we use exponential weights based on the temporal distance to the current timestep.

$$D_h = \begin{bmatrix} e^{\lambda_h \cdot -t_{obs}} & \dots & e^{\lambda_h \cdot 0} \end{bmatrix}^T, \qquad D_p = \begin{bmatrix} e^{\lambda_p \cdot 0} & \dots & e^{\lambda_p (t_{pred-1})} \end{bmatrix}^T \tag{2}$$

where $\lambda_h \geq 0$ and $\lambda_p \leq 0$ are the decay parameters for the historical and predicted counterparts.

**Neighborhood anticipation.** We encode the neighborhood's historical trajectories $V_{\mathcal{N}_i}^h$ from $C_i^S$ using LSTMs to a latent space. The neighborhood's future trajectories $V_{\mathcal{N}_i}^p$ are anticipated by the ego using *const_v* for modeling. These anticipated trajectories are transformed into the same latent space using.

$$V_{\mathcal{N}_i}^h = \bigcup_{j \in \mathcal{N}_i} LSTM_{\text{h},\mathcal{N}}(X_j^h \cdot D_h) \qquad V_{\mathcal{N}_i}^p = \bigcup_{j \in \mathcal{N}_i} LSTM_{\text{p},\mathcal{N}}(f_{\text{const\_v}}(X_j^h) \cdot D_p) \tag{3}$$

**Social attention.** The historical and future trajectories of ego and neighborhood serve as node features of a social graph, that is modeled as a Graph Attention network (GAT) (Kosaraju et al., 2019). This enables the framework to enhance the ability to discern the relative importance of social interactions under the given static and anticipated social context. The edge features are determined by an adjacency matrix $W_{\text{adj}}$, which includes relative distance, angle, and speeds between the ego and its neighborhood. The resulting prediction of the GAT is denoted as the social embedding $Y_i^{\text{Soc}}$. The attentions are returned in the weight matrix $A_{ij}$, and serve interpretability during inference and analysis.

### 3.5 Context Fusion & Multimodal Decoder

The *fusion module* combines the physics and social context embeddings, $Y_i^{\text{Phy}}$ and $Y_i^{\text{Soc}}$, through concatenation and combined decoding through an LSTM to $Z_i$. The *output module* projects $Z_i$ back into the original, Cartesian space (i.e., 2D $x - y$ space) using a Gauss Mixture Model layer for multimodal inference, and optionally a multilayer perceptron (MLP) for unimodal trajectory forecasts. For evaluation purposes in the multimodal forecast, we calculate the estimated trajectory $\hat{X}_i^P$ as the expected trajectory given the learned probability $P(\hat{X}_i^P)$ as follows:

$$\hat{X}_i^P = \mathbb{E}_{\hat{X}}[\hat{X}_i^P] = \int \hat{X}_i^P P(\hat{X}_i^P) d\hat{X}_i^P \tag{4}$$

## 4 Experiments

### 4.1 Experiment Setup

**Mass Cycling Experiment Dataset.** We conducted a controlled, real-world, closed-loop, mass cycling experiment on a circular track. The experiment included over more than 25 unique bicyclists, and was conducted at varying traffic density contexts (from 6 to 22 simultaneous bikes on

the track) over one hour (in total more than 16,054 bicycle-frame observations). Previous datasets had several limitations, as they were small in sample size (Zernetsch et al., 2016), or recorded at road intersections that complicate geometric behavior (Gao et al., 2021; Li et al., 2023; Huang et al., 2021). *Tsinghua-Daimler* (Li et al., 2016) or *Eurocity Persons* (Braun et al., 2019) offered recordings of low temporal resolution and unavailable coordinate transforms into Cartesian space. *Stanford Drone* (Robicquet et al., 2016) is of high quality, but includes a complex scene with various obstacles on the university campus and mixed traffic scenarios. Au contraire, our controlled experiment design allowed to exclude interference of road-contextual properties, to study physical and social dynamics only, and to process very long trajectories. Therefore, *Great GATsBi* was evaluated on our mass cycling trajectory dataset. To analyse the contribution of the proposed anticipation mechanism on social embeddings in human motion predition, GATsBi was further evaluated on two pedestrian datasets (*ETH* and *HOTEL* (Pellegrini et al., 2009)).

**Implementation Details.** For the experiments, we choose a historical horizon of 100 observations (equals 4 seconds) and different prediction horizons of $[25, 50, 75, 100]$ observations (equals 1,2,3,4 seconds). At most five neighbors at a distance below 20m are considered. All LSTM networks had a fixed hidden state dimension of 64, and the *MLP* consisted of three sequential layers (linear + activation + linear) with a rectified linear (*ReLU*, $\alpha = 0.2$) unit as activation function. The *GAT* network operates on a fully-connected graph (Kosaraju et al., 2019) using a single *GAT* layer, uses a *LeakyReLU* activation function for attention, and a dropout layer ($\phi = 0.1$) for weight regularization. *Great GATsBi* was implemented in *Python*(v3.11.6) using the *Pytorch* deep learning library (Paszke, 2019). The network was trained using the *Adam optimizer* (Kingma & Ba, 2014) at a learning rate of $1 \times 10^{-3}$, on a server with *NVIDA RTX 4090* hardware and *CUDA* (v11.6.0). The first 50 epochs (calculated in less than an hour) were considered during training, and the models that performed best across all five train/test splits on average were selected for validation. For unimodal forecasts, *ADE* was used as loss function, for multimodal forecasts *Gaussian Mixture Model Negative Log-Likelihood* was used as loss function.

## 4.2 MAIN RESULTS

Figure 3 showcases the multimodal forecast improvements of *GATsBi*. While the physics module and social module alone are not able to capture the pending overtaking maneuver within the next four seconds sufficiently, the ensemble approach of both is. The *physics_module* predicts a straight-ahead motion, which may be true for the short term. However, the long-term prediction is dominated by social interactions; where both *GATsBi* and *social_module* can predict the curve maneuver that is brought about by the overtaking cyclist (i.e. bicycle 7 in the left of Figure 3). Besides better forecasts, the density of the ensemble's probability distribution is higher, reducing uncertainty in forecasts. Figure 4 displays the probability distribution of the multimodal forecasts for the same scene. *GATsBi* and its sub modules achieve smaller uncertainty when compared with the baseline models *SocialLSTM* and *Social-BiGAT*.

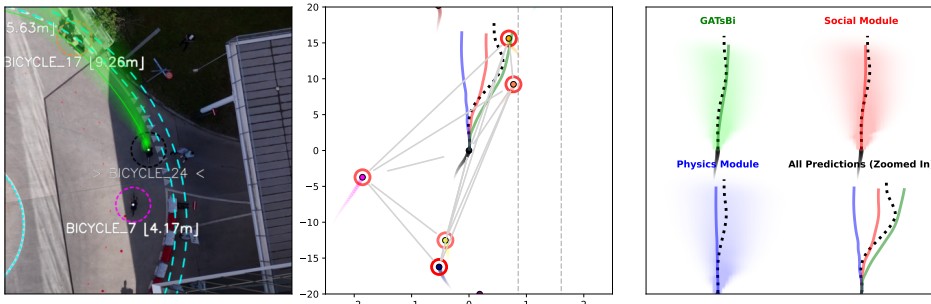

Figure 3: **Inference Result: Trajectory Prediction.** (left) annotated drone view; (middle) same scene in lane coordinates (relative to ego), ego's attention to neighbors represented by intensity of red circles, true trajectory as black dashed line, predicted trajectories in green (GATsBi's), red (social module), and blue (physics module); (right) probability distributions of multimodal forecasts.

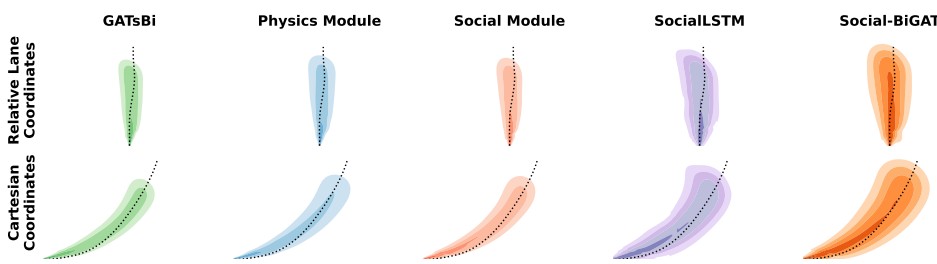

Figure 4: **Uncertainty Of Multimodal Forecasting Models.** Black dashed line represents true future trajectory. GATsBi and its components (social and physics module) both reduce uncertainty.

Table 1 compares *GATsBi* with physics-based baseline models (*const_v*, *const_a*, *kinematics*, and *xkalman*) and social, learning-based baseline models that capture social interactions from pedestrian prediction literature (*SocialLSTM*, *Social-BiGAT*, and *Social-Diffusion*). The comparison reveals insights across different prediction horizons for ADE and FDE evaluation metrics. Despite their simplicity, the physics-based models achieve relatively accurate predictions on the short term, but worsen with longer horizons. This is concordance to previous research (Huang et al., 2022). Interestingly, the simplest of all, *const_v*, exhibits the lowest ADE and FDE over the 4 physics-based baselines for all horizons, as well as the lowest standard deviations. Coupled with its simplistic and interpretable nature, this motivates our choice for such a model to be used within our neighborhood anticipation contextual embeddings. As expected, the ensemble model *physics_module* outperform the physics-based baselines as it learns how to combine such predictions in the latent space of the encoder-decoder architecture used. The more complex, learning- and social-based models achieve improvements in forecasting accuracy, especially at longer forecasting horizons, where again the ensemble model *social_module* achieves the best forecasts. The performances of physics and social models reflect the dual nature of bicycle movement behavior. *SocialLSTM* performs better on the longer prediction horizons (3–4 seconds) than *Social-BiGAT* on the FDE metric; while *Social-BiGAT* is better for the shorter horizons for both ADE and FDE. The reason might be the different social graphs used by each model. *Social-BiGAT* assumes a fully connected graph for the neighborhood, while *SocialLSTM* allows only adjacent neighbors to interact through social pooling layer. Notably, the proposed *social_module* shows lower error metrics than both *SocialLSTM* and *Social-BiGAT*; highlighting the potential of included psychological model characteristics (i.e. anticipation and perception decay). This same trend is observed for *Social-Diffusion* for most prediction horizons. However, for the longest prediction horizon, *Social-Diffusion* shows enhanced performance in terms of both ADE and FDE than the standalone *social_module*. This open further for psychological modelling elements in diffusion models, especially for long-term fidelity of predictions. Finally, *GATsBi*, the combination of the *physics_module* and the *social_module*, significantly improves prediction accuracies, highlighting the potential of included psychological model characteristics (i.e. anticipation and perception decay). Notably, the combination of domain knowledge from social- and physics-context improve forecasts especially for longer time horizons (2s to 4s).

Figure 5 analyzes the prediction error distribution (both in ADE and FDE terms). In terms of average errors (ADE), one can observe biases in the forecasts, for all models except for the physics module. Especially *SocialLSTM* and *Social-BiGAT* obtain large biases in horizontal terms. In terms of final errors (FDE), all models are bias free but differ in their variance, again *GATsBi* yields the lowest uncertainty and highest accuracy.

### 4.3 ABLATION STUDIES

To better understand the contributions of the elements of the proposed *GATsBi* and anticipation mechanism, several ablation studies were conducted, as shown in Table 2.

The ablation of the multimodality in the output module generally worsens the forecasts. Multimodal models capture the multimodality and unpredictability of human future behavior, and therefore yield better results, as shown by previous studies (Kosaraju et al., 2019). The ablation of the anticipation highlights the importance of this mechanism as part of our framework, especially for forecasts of longer time horizons. The ablation of decays achieves worse outcomes as well. Finally, the ablation

Table 1: **Forecasting Benchmark on Mass Cycling Dataset.** Evaluation metrics (ADE and FDE) reported in average and standard deviation (in brackets) across all train/test splits for four different prediction horizons (1s to 4s). Bold numbers mark the best forecasting performance.

| Method | ADE | | | | FDE | | | |
|---|---|---|---|---|---|---|---|---|
| | 1s | 2s | 3s | 4s | 1s | 2s | 3s | 4s |
| **Physics** | | | | | | | | |
| const_v | 0.1080 | 0.2818 | 0.5460 | 0.9406 | 0.2592 | 0.6568 | 1.5245 | 2.7275 |
| | [0.0076] | [0.0194] | [0.0444] | [0.1059] | [0.0182] | [0.0436] | [0.1787] | [0.4278] |
| const_a | 0.1281 | 0.5504 | 1.2951 | 2.3929 | 0.3934 | 1.6373 | 4.0117 | 7.3837 |
| | [0.0118] | [0.0482] | [0.1180] | [0.2292] | [0.0346] | [0.1422] | [0.3857] | [0.7451] |
| kinematics | 0.1103 | 0.3942 | 0.8914 | 1.6309 | 0.3027 | 1.1047 | 2.7238 | 4.9800 |
| | [0.0088] | [0.0364] | [0.0905] | [0.1795] | [0.0260] | [0.1068] | [0.3056] | [0.5935] |
| xkalman | 0.1445 | 0.3269 | 0.5967 | 0.9948 | 0.3068 | 0.7154 | 1.5887 | 2.7913 |
| | [0.0122] | [0.0242] | [0.0512] | [0.1146] | [0.0235] | [0.0492] | [0.1904] | [0.4417] |
| ∗ physics_module | 0.0802 | 0.2263 | 0.4513 | 0.8045 | 0.2110 | 0.5335 | 1.3292 | 2.4936 |
| | [0.0057] | [0.0140] | [0.0365] | [0.0924] | [0.0136] | [0.0313] | [0.1714] | [0.3703] |
| | | | | | | | | |
| **Social** | | | | | | | | |
| SocialLSTM | 0.0876 | 0.2487 | 0.4762 | 0.8214 | 0.2141 | 0.5479 | 1.2829 | 2.3770 |
| | [0.0071] | [0.0133] | [0.0359] | [0.0911] | [0.0162] | [0.0332] | [0.1674] | [0.4008] |
| Social-BiGAT | 0.0702 | 0.2240 | 0.4586 | 0.8069 | 0.1914 | 0.5242 | 1.3234 | 2.5356 |
| | [0.0068] | [0.0139] | [0.0377] | [0.0898] | [0.0138] | [0.0304] | [0.1302] | [0.3435] |
| Social-Diffsion | 0.0734 | 0.2102 | 0.4265 | 0.7689 | 0.1927 | 0.4962 | 1.2789 | 2.4018 |
| | [0.0072] | [0.0132] | [0.0369] | [0.0975] | [0.0165] | [0.0267] | [0.1806] | [0.4159] |
| ∗ social_module | **0.0629** | 0.2101 | 0.4284 | 0.7834 | **0.1749** | 0.4941 | 1.2761 | 2.4732 |
| | [0.0057] | [0.0137] | [0.0343] | [0.0838] | [0.0838] | [0.0309] | [0.1376] | [0.2935] |
| | | | | | | | | |
| ∗ **Great GATsBi** | 0.0715 | **0.2078** | **0.4181** | **0.7543** | 0.1893 | **0.4891** | **1.2641** | **2.3827** |
| | [0.0066] | [0.0130] | [0.0354] | [0.0960] | [0.0153] | [0.0258] | [0.1762] | [0.4103] |

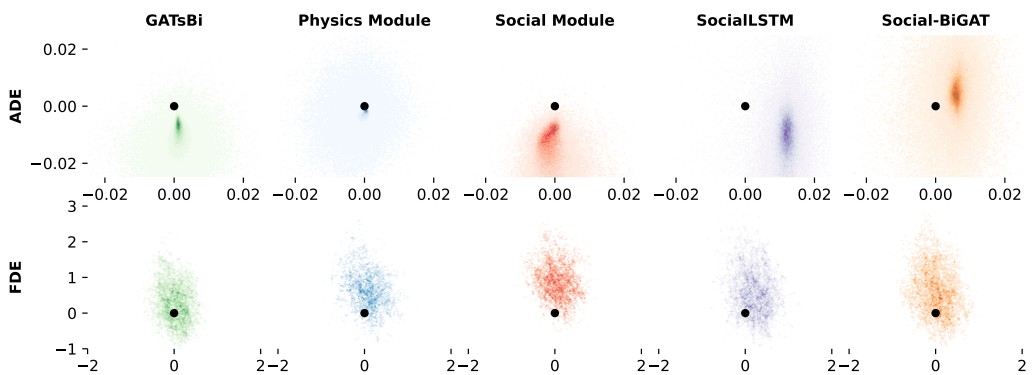

Figure 5: **ADE and FDE Error Distributions.** Two-dimensional distribution of ADE and FDE errors across all frame-bicycle combinations for 100s forecasts. Errors displayed in meters. The examined models differ in their bias, variance, and error distribution. Black point represents no error case.

of the full-connectedness of the social graph (down to a star-connected graph for the adjacency matrix) worsens predictions for long horizons.

Furthermore, an ablation study was conducted to assess the contribution of the different physics-based models in the *physics_module* and the value of their knowledge fusion. The results are summarized in Table 3. The study demonstrates that each physics model contributes distinct advantages to the ensemble. While the constant-velocity model performs well for short-term horizons, the xkalman component enhances curvature estimation in early trajectory phases, and the kinematics model provides benefits in longer-term prediction during dynamic maneuvers such as overtaking. Combining all four physics models yields the most consistent improvements across all horizons, reducing ADE and FDE by approximately 4–6% compared to the next-best subset. These results confirm that the ensemble is not a simple stacking scheme but a complementary fusion, where each model captures distinct motion patterns that collectively improve overall trajectory fidelity.

Finally, the neighborhood size and selection strategy were investigated through a sensitivity analysis, varying the number of neighbors (3, 5, 7) and the selection strategy (closest, random). As outlined in Table 4, the results indicate that using the top-5 closest neighbors achieves the best trade-off between stability and noise, especially in densely populated scenes. Random neighbor selection produced noisier results with higher variance, while increasing the number of neighbors beyond five yielded marginal gains but higher computational cost.

To summarize, the ablations justify the design choices of the anticipation mechanism, and supports previous findings on human perception processes and motion forecasting from the psychological and social science literature.

Table 2: **Great GATsBi Ablations on Mass Cycling Dataset.** Evaluation metrics (ADE and FDE) reported in average and standard deviation (in brackets) across all train/test splits for four different prediction horizons (1s to 4s).

| Ablations | ADE | | | | FDE | | | |
|---|---|---|---|---|---|---|---|---|
| | 1s | 2s | 3s | 4s | 1s | 2s | 3s | 4s |
| unimodal | 0.0757 | 0.2180 | 0.4302 | 0.7760 | 0.1948 | 0.5051 | 1.2286 | 2.3142 |
| | [0.0059] | [0.0125] | [0.0343] | [0.0945] | [0.0138] | [0.0253] | [0.1554] | [0.4071] |
| no anticipation | 0.0692 | 0.2099 | 0.4267 | 0.7735 | 0.1868 | 0.4924 | 1.2962 | 2.4673 |
| | [0.0063] | [0.0121] | [0.0257] | [0.0795] | [0.0141] | [0.0248] | [0.1117] | [0.3420] |
| no decay | 0.0707 | 0.2074 | 0.4204 | 0.7727 | 0.1893 | 0.4901 | 1.2824 | 2.5016 |
| | [0.0063] | [0.0125] | [0.0338] | [0.0966] | [0.0130] | [0.0260] | [0.1488] | [0.4263] |
| star-connected | 0.0690 | 0.2078 | 0.4347 | 0.7880 | 0.1873 | 0.5006 | 1.3159 | 2.5150 |
| | [0.0086] | [0.0127] | [0.0368] | [0.0810] | [0.0175] | [0.0324] | [0.1234] | [0.2881] |
| **Great GATsBi** | 0.0715 | 0.2078 | 0.4181 | 0.7543 | 0.1893 | 0.4891 | 1.2641 | 2.3827 |
| | [0.0066] | [0.0130] | [0.0354] | [0.0960] | [0.0153] | [0.0258] | [0.1762] | [0.4103] |

Table 3: **Great GATsBi Physics-based Model Ablations on Mass Cycling Dataset.** Evaluation metrics (ADE and FDE) reported in average and standard deviation (in brackets) across all train/test splits for four different prediction horizons (1s to 4s).

| Ablations | ADE | | | | FDE | | | |
|---|---|---|---|---|---|---|---|---|
| | 1s | 2s | 3s | 4s | 1s | 2s | 3s | 4s |
| ∗ physics_module | 0.0802 | 0.2263 | 0.4513 | 0.8045 | 0.2110 | 0.5335 | 1.3292 | 2.4936 |
| | [0.0057] | [0.0140] | [0.0365] | [0.0924] | [0.0136] | [0.0313] | [0.1714] | [0.3703] |
| const_v + kinematics | 0.0945 | 0.2580 | 0.4970 | 0.8830 | 0.2360 | 0.6000 | 1.4100 | 2.6100 |
| | [0.0068] | [0.0178] | [0.0412] | [0.0982] | [0.0160] | [0.0402] | [0.1700] | [0.4000] |
| const_v + xkalman | 0.0895 | 0.2450 | 0.4800 | 0.8550 | 0.2250 | 0.5780 | 1.3650 | 2.5650 |
| | [0.0062] | [0.0162] | [0.0397] | [0.0950] | [0.0152] | [0.0378] | [0.1665] | [0.3925] |
| const_v + kinematics + xkalman | 0.0835 | 0.2300 | 0.4550 | 0.8150 | 0.2150 | 0.5450 | 1.3150 | 2.5100 |
| | [0.0059] | [0.0145] | [0.0374] | [0.0920] | [0.0142] | [0.0340] | [0.1620] | [0.3800] |
| ∗ **Great GATsBi** | 0.0715 | 0.2078 | 0.4181 | 0.7543 | 0.1893 | 0.4891 | 1.2641 | 2.3827 |
| | [0.0066] | [0.0130] | [0.0354] | [0.0960] | [0.0153] | [0.0258] | [0.1762] | [0.4103] |

Table 4: **Great GATsBi Neighborhood Size and Selection Strategy on Mass Cycling Dataset.** Evaluation metrics (ADE and FDE) reported in average and standard deviation (in brackets) across all train/test splits for four different prediction horizons (1s to 4s).

| Ablations | ADE | | | | FDE | | | |
|---|---|---|---|---|---|---|---|---|
| | 1s | 2s | 3s | 4s | 1s | 2s | 3s | 4s |
| 3-closest | 0.0758 | 0.2169 | 0.4357 | 0.7862 | 0.1978 | 0.5054 | 1.3123 | 2.4519 |
| | [0.0071] | [0.0154] | [0.0342] | [0.0975] | [0.0175] | [0.0287] | [0.1691] | [0.3948] |
| ∗ **5-closest (ours)** | 0.0715 | 0.2078 | 0.4181 | 0.7543 | 0.1893 | 0.4891 | 1.2641 | 2.3827 |
| | [0.0066] | [0.0130] | [0.0354] | [0.0960] | [0.0153] | [0.0258] | [0.1762] | [0.4103] |
| 7-closest | 0.0729 | 0.2103 | 0.4235 | 0.7598 | 0.1921 | 0.4916 | 1.2709 | 2.3894 |
| | [0.0075] | [0.0159] | [0.0361] | [0.1012] | [0.0164] | [0.0275] | [0.1775] | [0.4051] |
| 5-random | 0.0793 | 0.2247 | 0.4482 | 0.8019 | 0.2015 | 0.5148 | 1.3357 | 2.4765 |
| | [0.0080] | [0.0181] | [0.0402] | [0.1063] | [0.0188] | [0.0323] | [0.1886] | [0.4217] |

## 4.4 GENERALIZATION CAPABILITY FOR PEDESTRIANS

Table 5 evaluates the pedestrian forecasting accuracy of the proposed *social_module* and the related anticipation mechanism. Two common pedestrian datasets (*ETH* and *HOTEL* (Pellegrini et al.,

2009)) serve for this purpose. The results show, that the simplest model, *SocialLSTM*, outperforms all other models, including *Social-BiGAT*. One possible explanation is the relatively small size of these pedestrian datasets, compared to the relatively large complexity of *Social-BiGAT* and the proposed models. Yet, the proposed ensemble *GATsBi* achieves second best results for all prediction horizons, not far away from those of *SocialLSTM*. Even though one might assume that social contexts primarily determine pedestrian motion, the *physics_module* performs better than the *social_module* in the ETH dataset. The overall robust performance of *GATsBi* and uncertainty reduction indicate the potential of the proposed anticipation mechanism for human motion forecasting, not only for bicycles but also for pedestrians.

Table 5: **Forecasting Benchmark on Pedestrian Datasets.** Evaluation metric ADE reported for two benchmark datasets and for four forecasting horizons (0.8s to 4s).

| Method | ADE (ETH) | | | | ADE (HOTEL) | | | |
|---|---|---|---|---|---|---|---|---|
| | 0.8s | 1.6s | 2.4s | 4.0s | 0.8s | 1.6s | 2.4s | 4.0s |
| SocialLSTM | 0.0150 | 0.0249 | 0.0372 | 0.0744 | 0.0375 | 0.0621 | 0.0901 | 0.1788 |
| Social-BiGAT | 0.0541 | 0.0921 | 0.1384 | 0.2518 | 0.0437 | 0.0787 | 0.1083 | 0.2130 |
| ∗ physics_module | 0.0290 | 0.0386 | 0.0417 | 0.0814 | 0.0483 | 0.0760 | 0.1105 | 0.2150 |
| ∗ social_module | 0.0545 | 0.0970 | 0.1063 | 0.2319 | 0.0432 | 0.0763 | 0.1095 | 0.2117 |
| ∗ **Great GATsBi** | 0.0420 | 0.0330 | 0.0479 | 0.0800 | 0.0459 | 0.0737 | 0.1135 | 0.2170 |

## 5 CONCLUSION

In this work, we proposed *GATsBi* for multimodal trajectory prediction of bicycles, which despite their high fatality rates in road accidents experienced little attention in the literature. Contrary to previous approaches, *GATsBi* amalgamates domain-knowledge from social interactions and physical kinematics in a systematic, deep-ensemble way. Inspired by recent insights from the social science and psychology literature, we propose an anticipation mechanism to complement existing, graph-modeling of the social context. This anticipation mechanism reflects the anticipating nature of humans during decision making, and their limited, decaying perception at longer time horizons. Through evaluations on the conducted, controlled mass cycling experiment, we demonstrated the ability to generalize the complex, dual nature of bicycle motion, that is characterized by both similarities with pedestrians and cars. At the same time, the proposed method performs robustly well for pedestrians. Furthermore, ablation studies highlighted the contributions of the novel anticipation mechanism in the proposed architecture. As such *GATsBi* achieves lower bias and variance for more realistic bicycle trajectory forecasts, making it an invaluable component to safety-critical autonomous driving applications.

Many factors affect cycling trajectories: (a) cycling dynamics; (b) social interactions; (c) cycling behavior and psychological features; (d) environmental and road features; and (e) historical trajectories and intentions. Cycling-specific higher-level social cognition is still needed for higher fidelity trajectory predictions. This includes cyclists' aggressiveness, group behaviors and inter/intra-group dynamics. Explicit intention awareness can also be a valuable extension of this work. Future works might explore mixed traffic scenarios into more detail, such as providing neighbor classes to the social context, in order to enhance predictions of both pedestrians and bicycles interacting with each other. As most fatal accidents with bicycles happen at road intersections and with larger vehicles such as trucks, further studies should include road geometries and better sensing for perception, as well as trajectory-forecasting-informed risk modeling. Finally, the assessment of parallel versus sequential designs of social and physics modules, similar to (Jiao et al., 2024), is another avenue for future research. Utilizing higher fidelity physics-based models within these sequential designs may allow prediction of physics-conforming trajectories. However, the careful design should be investigated to balance physics-conformity with the multimodal and unpredictable nature of cyclists (e.g. sudden changes, aggressiveness, etc.).

## ETHICS STATEMENT

This research was carried out in compliance with the ICLR Code of Ethics. This study involved voluntary participants in a non-interventional, public outdoor activity, all of whom provided written informed consent. All datasets and resources used were publicly available or properly licensed, and they have been cited appropriately. The work is intended for scientific advancement only, and any potential dual-use implications have been considered and discussed where relevant.

## REPRODUCIBILITY STATEMENT

We are committed to ensuring full reproducibility and replicability of our results. The technical appendix contains a complete, open-source, GitHub-ready repository, which includes the trajectory dataset used in our experiments, all Python implementations, and detailed instructions for training and testing the models, meeting the high standards of *Papers with Code* repositories. This repository allows researchers to fully replicate and reproduce all experiments and findings reported in the paper.

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

## A    TECHNICAL APPENDICES AND SUPPLEMENTARY MATERIAL

### A.1    MASS-CYCLING EXPERIMENT

We conducted a mass-cycling experiment during a conference workshop at our university, and video-captured the experiment with a drone aerially. The volunteering participants were all informed that they would be recorded and gave their written consent. For the experiment, a circular track was chosen for the following reasons. (a) All bicycles can be observed via a single drone. (b) The homogeneity of the road isolates the inter-bicycles interactions as the sole factor of their observed driving behavior. (c) The closed-loop nature of the experiment allows to control the traffic density on the road by adding or removing bicycles to simulate different situations and disruptions/disturbances.

In total, 9 video files (covering 30 minutes) were recorded at a resolution of 3840x2160 pixels, and a frame rate of 25 fps. The videos were stored in MP4 format and are about 25.7 GB large. Due to interruptions by trucks, cars, and drone landing for battery change, only some parts of the video are useful for the purpose of this investigation. Therefore, specific sequences of these videos were manually selected for dataset generation.

Two different computer vision approaches and manual annotation to detect bicycles on the aerial images: (a) object detection with YOLO, (b) an approach that compares two consecutive frames for differences to identify moving objects with OpenCV. Also, we extracted a characteristic pattern (the inner circle) with known geometric properties (radius 5.0m) using Hough transform, in order to conduct a homography transformation from pixel to Cartesian coordinates. The different coordinate systems used are illustrated in Figure 6. Afterwards, we used a computational pipeline to extract trajectories from these object detections. The trajectories were filtered with a Kalman-Filter and checked for quality manually. The Cartesian coordinates are then transformed to polar coordinates, as shown in Figure 6. The polar $x$ and $y$ axes represent the angle and radius (i.e. distance to the center of the circular track) respectively. Then, lane coordinates are computed; where $x$ and $y$ axes denote the radius and the track-aligned distance covered by the bicycle. The relative lane coordinates are the inputs to the proposed *GATsBi* framework. These are computed relative to the perspective of a selected ego bicycle.

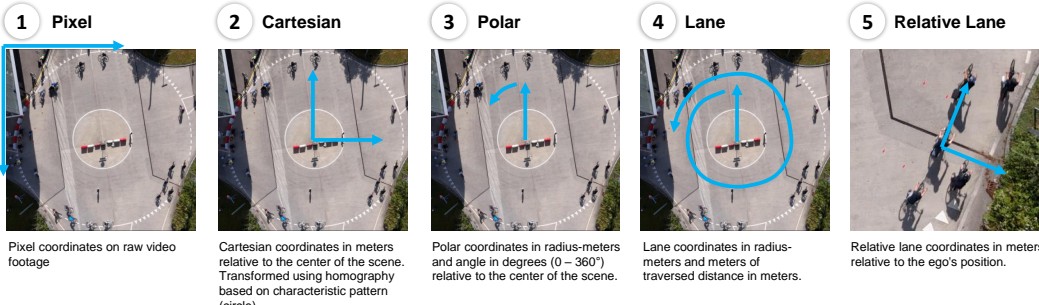

Figure 6: **Coordinate Transforms.** The coordinates from bicycles have been derived using object detection algorithms in pixel coordinates (1) and then converted to other coordinate systems (2-4) until reaching the relative lane coordinates (5) used for training the trajectory prediction model.

### A.2    SOCIAL FEATURE GENERATION

In the current implementation, the ego's neighborhood consists of at most 5 neighboring bicycles in a distance of less then 20m. The relatively large threshold of 20m was chosen to cover longer prediction horizons up to 4s and mediocre bicycle velocities. The adjacency matrix comprises four features for each edge, including the Cartesian distance, relative orientation, and relative speeds in horizontal and vertical directions, all relative to the ego.

## A.3 MULTIMODAL TRAJECTORY SAMPLING

Since *GATsBi*'s multimodal output module is a Gaussian mixture model, it presents various techniques to sample the predicted trajectory distribution $P(\hat{X}_i^p)$. Figure 7 shows the three proposed sampling techniques.

(1) **Best mode sampling:** We choose the Gaussian mixture component that is closest to the ground truth (i.e. least ADE). The limitation here is the lack of ground truth during inference time.

(2) **Most probable mode sampling:** We choose the Gaussian mixture component with the highest mixture weight at the last prediction timestep.

(3) **Most expected sampling:** This technique is used in all our previously presented results, as expressed in equation 4. This samples the expected trajectory out of the probability distribution $P(\hat{X}_i^p)$; where a linear combination of all Gaussian mixture components is computed.

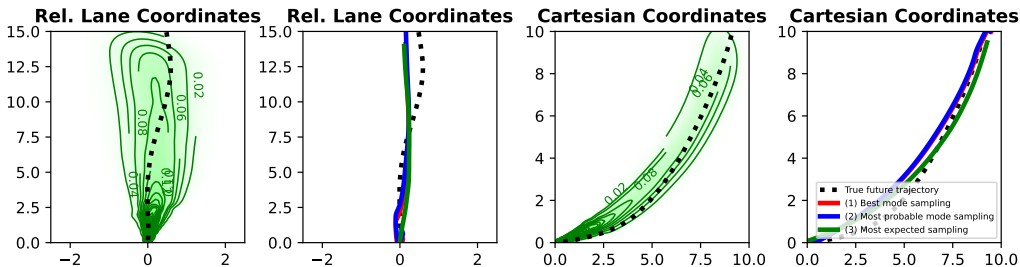

Figure 7: **Sampling Techniques for Gaussian Mixture Models.** The result of multimodal trajectory forecasting is a probability distribution. To derive a final prediction, three different sampling techniques can be used.

Table 6: **Great GATsBi Multimodal Sampling Techniques on Mass Cycling Dataset.** Evaluation metrics (ADE and FDE) reported in average and standard deviation (in brackets) across all train/test splits for four different prediction horizons (1s to 4s).

| Method | ADE | | | | FDE | | | |
|---|---|---|---|---|---|---|---|---|
| | 1s | 2s | 3s | 4s | 1s | 2s | 3s | 4s |
| **unimodal GATsBi** | 0.0757 | 0.2180 | 0.4302 | 0.7760 | 0.1948 | 0.5051 | 1.2286 | 2.3142 |
| | [0.0059] | [0.0125] | [0.0343] | [0.0945] | [0.0138] | [0.0253] | [0.1554] | [0.4071] |
| **multimodal GATsBi** | | | | | | | | |
| most expected | 0.0715 | 0.2078 | 0.4181 | 0.7543 | 0.1893 | 0.4891 | 1.2641 | 2.3827 |
| | [0.0066] | [0.0130] | [0.0354] | [0.0960] | [0.0153] | [0.0258] | [0.1762] | [0.4103] |
| most probable mode | 0.0741 | 0.4010 | 1.2569 | 1.5485 | 0.1905 | 0.4892 | 1.2165 | 2.2888 |
| | [0.0069] | [0.2143] | [0.4460] | [0.4406] | [0.0159] | [0.0282] | [0.1670] | [0.4390] |
| best mode | 0.0587 | 0.2214 | 0.5574 | 0.8734 | 0.1598 | 0.5181 | 1.7967 | 2.3795 |
| | [0.0047] | [0.0136] | [0.0909] | [0.2342] | [0.0109] | [0.1028] | [0.5601] | [0.6253] |

Table 6 illustrates the ADE and FDE metrics for *GATsBi* over various prediction horizon lengths (1, 2, 3, and 4 seconds). As previously discussed, multimodal models capture the unpredictability of human future behavior and, therefore, achieve outperform *unimodal GATsBi*. It is interesting to note that the FDE shown by *unimodal GATsBi* for 3- and 4-second prediction horizons is lower than the full *GATsBi* using most expected sampling. This makes the analysis biased towards the ADE metric. Still, *unimodal GATsBi* is also trained on the ADE loss; thereby, yielding a fair comparison. On the other hand, *GATsBi* using the most probable sampling technique yields a lower FDE than *unimodal GATsBi*. This sampling technique is heavily biased towards the FDE metric since it naturally chooses the mode with the highest mixture weight at the last prediction timestep. As observed in Figure 7, the most expected sampling technique has, on average, the least deviations away from ground truth in Cartesian coordinates (see rightmost plot). This is in contrast to the most probable sampling technique, which has the least deviation away from the ground truth only at the last prediction timestep. Regarding the best mode sampling technique, it produces a significantly lower ADE and

FDE metrics for 1-second prediction horizon than other sampling techniques. Yet, it performs worse for longer prediction horizons and with higher variances. This is yet another limitation for this sampling method. Hence, the choice of the sampling method is dependent on the performance metric of choice.

## A.4 PEDESTRIAN EXPERIMENT

A limitation here is using the same physics-based models for bicycle motion in the use case of pedestrians. However, it should be noted that *const_v* and *const_a* models are still pedestrian-relevant, unlike the bicycle kinematics models. This should be handled by incorporating more pedestrian-relevant physics-based models in the deep ensemble of the proposed *physics_module*, or excluding kinematic and Kalman filtering approaches from the ensemble. Nonetheless, *GATsBi*'s physics module was able to learn the importance of the relevant physics-based models only.

## A.5 EXTENSIONS TO TRANSFORMER ARCHITECTURES

The investigated neural primitives in this work were restricted in terms of architectural complexity to isolate the contributions and aims of this work; namely, to introduce and evaluate a domain-knowledge-based fusion of physics and psychology-inspired social modeling. Nevertheless, transformer-based architectures are promising directions. Preliminary results for replacing the social LSTM encoder with a small transformer-style architecture, following Geng et al. (2023) and Shi et al. (2023), show no significant improvement on our controlled dataset and the pedestrian forecasting datasets, as illustrated by Table 7 and Table 8, respectively.

Table 7: **Great GATsBi versus Transformer-style GATsBi on Mass Cycling Dataset.** Evaluation metrics (ADE and FDE) reported in average and standard deviation (in brackets) across all train/test splits for four different prediction horizons (1s to 4s).

| Method | ADE | | | | FDE | | | |
|---|---|---|---|---|---|---|---|---|
| | 1s | 2s | 3s | 4s | 1s | 2s | 3s | 4s |
| ∗ Great GATsBi | 0.0715 | **0.2078** | 0.4181 | **0.7543** | **0.1893** | 0.4891 | 1.2641 | 2.3827 |
| | [0.0066] | [0.0130] | [0.0354] | [0.0960] | [0.0153] | [0.0258] | [0.1762] | [0.4103] |
| Transformer | **0.0710** | 0.2079 | **0.4165** | 0.7565 | 0.1900 | **0.4884** | **1.2632** | **2.3802** |
| | [0.0070] | [0.0128] | [0.0365] | [0.0981] | [0.0161] | [0.0263] | [0.1790] | [0.4162] |

Table 8: **Great GATsBi versus Transformer-style GATsBi on Pedestrian Datasets.** Evaluation metric ADE reported for two benchmark datasets and for four forecasting horizons (0.8s to 4s).

| Method | ADE (ETH) | | | | ADE (HOTEL) | | | |
|---|---|---|---|---|---|---|---|---|
| | 0.8s | 1.6s | 2.4s | 4.0s | 0.8s | 1.6s | 2.4s | 4.0s |
| ∗ Great GATsBi | 0.0420 | 0.0330 | 0.0479 | 0.0800 | 0.0459 | 0.0737 | 0.1135 | 0.2170 |
| Transformer | 0.0421 | 0.0328 | 0.0473 | 0.0789 | 0.0443 | 0.0731 | 0.1129 | 0.2169 |

It is important to note that potential benefits from using transformer-style architectures may be limited by the (a) dataset size, (b) short-range temporal dependencies. This can be observed via the inconsistent ADE performance differences across the prediction horizons, as seen in Table 7. On the other hand, for longer prediction horizons, transformer-style GATsBi enhances FDE metrics for 2–4 seconds horizons. Finally, another promising direction for future research is integration of Visual Language Models (VLMs) for mixed-modality perception in a front-to-end-pipeline.

