# OpenReview forum: "Great GATsBi: Hybrid, Multimodal, Trajectory Forecasting for Bicycles using Anticipation Mechanism"
_ICLR.cc/2026/Conference — Submitted to ICLR 2026_

### Official Review · Reviewer_EYab · 2025-10-29

**Soundness:** 2
**Presentation:** 2
**Contribution:** 2
**Rating:** 4
**Confidence:** 5

**Summary:**

The paper introduces Great GATsBi, a hybrid multimodal framework for bicycle trajectory prediction that combines physical dynamics and social interaction modeling. The physics module includes constant velocity, constant acceleration, kinematic, and Kalman filter models, encoded through LSTMs. The social module integrates neighbor trajectories through GATs to model inter-agent interactions. Evaluated on a multi-cyclist dataset collected from the authors' controlled real-world experiment, Great GATsBi outperforms Social-LSTM and Social-BiGAT in ADE/FDE over 1s~4s horizons and demonstrates modest generalization on pedestrian datasets ETH and HOTEL.

**Strengths:**

The paper introduces anticipation and perception decay mechanisms in social interaction modeling, drawing on psychological principles of human decision-making to explain cyclists’ characteristics of “limited memory and foresight” when predicting neighbors’ behaviors.

The authors collected a controlled real-world cycling dataset containing multi-cyclist interaction behaviors, which effectively validates the model’s capability to represent such scenarios.

**Weaknesses:**

1. The paper oversimplifies the problem by ignoring road and environmental context, which limits its real-world applicability.
2. The model architecture is simple and largely follows existing pedestrian trajectory prediction frameworks; it remains within the standard GAT + LSTM paradigm with little technical novelty.
3. The chosen physical dynamics models are generic for all vehicles and cannot adequately capture the unique dynamics of bicycles.
4. The social interaction modeling remains oversimplified, especially under dense or complex traffic conditions. Although the authors emphasize that the main challenges of pedestrian and cyclist trajectory prediction stems from sudden trajectory changes, the current model still relies solely on historical trajectories, neighbor anticipation, and perception decay, while neglecting richer and more dynamic social behaviors [1]. I am not sure whether only these two mechanisms are sufficient to address above key challenge in bicycle trajectory prediction.

5. (1) The experimental setup is closed and limited. It lacks mixed-traffic scenarios involving vehicles, whose psychological and physical influence on cyclists could be crucial but is ignored. (2) The uncertainty evaluation only shows a single trajectory and does not convincingly demonstrate the model’s generative advantages. (3)The authors claim strong long-term prediction performance, but the actual prediction horizons (1–4 s) is too short to support this claim.

[1] Martín-López IM, García-Taibo O, Aguiló A, Borràs PA. Understanding Factors Influencing Cycling Behaviour Among University Students and Staff: A Cross-Sectional Study. Social Sciences. 2025; 14(5):261. https://doi.org/10.3390/socsci14050261

**Questions:**

1. What challenges do the authors identify for bicycle trajectory prediction compared to pedestrian trajectory prediction?

2. What technical innovations does the GATsBi introduce compared to existing pedestrian trajectory prediction research?

3. How are the hyperparameters $\lambda_h$ and $\lambda_p$ set in the perception decay module?

---

> ### Author Response · Authors · 2025-11-20
> **Response to Reviewer EYab (Part 1/2)**
>
> We thank the reviewer for the assessment.
>
> **W1: “Ignoring road/environment limits real-world applicability.”**
>
> **W5: “Closed setup, modest uncertainty evaluation, horizon only up to 4s.”**
>
> We agree. Our controlled environment was deliberately chosen to isolate social and physical interactions without confounding from road geometry – as this is the focus of our work. We now clarify this limitation in the paper and outline ongoing work:
> 1. running GATsBi on SDD bicyclists (in progress),
> 2. extending neighbor classes (pedestrian/car) as suggested,
> 3. incorporating road geometry in the social module.
>
> While the transferability is part of our future work, we want to emphasize that it is common sense in the trajectory forecasting literature to work on circular ring roads to control for as many aspects as possible when working on vehicle trajectory forecasts: [Wu et al. 2022, Abouelazm et  al. 2025, Satish et al. 2024, Wang et al. 2025, Riehl et al. 2025].
> We respectfully disagree with the reviewer on this point of long-termity. In the literature on trajectory forecasting, 4s is considered long already. If you compare benchmark datasets such as Stanford Drone Dataset or ETH/HOTEL dataset, it is quite common to forecast up to 4s (and not beyond).
>
> **W2: “Architecture mostly standard; limited novelty.”**
>
> Our main novelty lies not in new neural primitives but in:
> - Hybrid deep-ensemble physics embedding + psychology-inspired social embeddings,
> - Anticipation + decay, shown to be effective through ablations,
> - New controlled bicycle dataset designed to isolate social/physical effects,
> - Interpretability through GAT attention and physics/social disentanglement.
> - Providing a transparent and reproducible code and benchmark for bicycle trajectory prediction task.
> We have clarified this contribution more explicitly.
>
> **W3: “Physics models generic, not bicycle-specific.”**
>
> We agree that bicycle-specific dynamics (nonholonomic constraints, lean angle) could be incorporated. However, the dataset is 2D aerial, making lean angle unobservable. We will note this limitation and discuss richer physics models as future work.
>
> **W4: “Social modeling oversimplified for dense traffic.”**
>
> Our controlled density ranges up to 22 cyclists; however, we agree urban scenes are more complex. The scope of our work is modelling cycling trajectories not cycling mode choice. Still, many factors affect cycling trajectories and they include: cycling dynamics (captured by the physics module); social interactions (captured by the social module); cycling behavior and psychological features (anticipation and perception decay); environmental and road features (extension for future work); and historical trajectories and intentions.
>
> In the revised manuscript, we:
> - clarify that anticipation/decay do not aim to fully model higher-level social cognition; for example, modelling cyclists’ aggressiveness, group behaviors and inter/intra-group dynamics.
> - discuss that historical trajectories can represent intention; however, explicit intention awareness can be a valuable extension of this work.
> It is important to note that our work shows the importance of fusion of domain knowledge with psychological modelling to enhance the performance of bicycle trajectory prediction models.

---

> ### Author Response · Authors · 2025-11-20
> **Response to Reviewer EYab (Part 2/2)**
>
> **Q1: What challenges do the authors identify for bicycle trajectory prediction compared to pedestrian trajectory prediction?**
>
> We identify following challenges:
> - nonholonomic constraints (bicycles cannot move sidewards)
> - higher speeds (compared to pedestrians)
> - overtaking (contrary to pedestrians)
> - flexible lane positioning (while no specific lane concept must exist for pedestrians)
>
> We add this explicitly in the revised version of the manuscript.
>
> **Q2: What technical innovations does the GATsBi introduce compared to existing pedestrian trajectory prediction research?**
>
> Our paper’s aim is to highlight that, while lots of research for pedestrian and car trajectory forecasting exists, bicycles are under-researched (even though they cause most of fatal accidents). It is explicitly not our aim to improve pedestrian trajectory forecasting.
> In our work, we contribute not only with better forecasting accuracy, but also with more transparency and explainability, and provide three core innovations:
> - Hybrid deep-ensemble physics embedding + psychology-inspired social embeddings,
> - Anticipation + decay, shown to be effective through ablations,
> - New controlled bicycle dataset designed to isolate social/physical effects,
> - Interpretability through GAT attention and physics/social disentanglement.
>
> **Q3: How are the hyperparameters and set in the perception decay module?**
>
> These parameters are learned during the training process (as attention parameters), as you can see in the attached code alongside our paper (e.g. in model_gatsbi.py):
> ```
>         # time decay
>         decay_vec = torch.arange(-T+1, 1).to(device=t_ego_hist.device) * self.history_dt
>         decay_vec = torch.exp(decay_vec * F.softplus(self.history_decay_param))
>         decay_vec = decay_vec.unsqueeze(0).unsqueeze(-1).repeat(B, 1, Ni)
>
>         # Encode ego
>         _, (h_ego, _) = self.agent_encoder(t_ego_hist * decay_vec)  # [1, B, hidden_dim]
>         h_ego = h_ego.squeeze(0)                      # [B, hidden_dim]
> ```
>
> **References :**
>
> - Abouelazm, Ahmed, et al. "Boundary-Guided Trajectory Prediction for Road Aware and Physically Feasible Autonomous Driving." arXiv preprint arXiv:2505.06740 (2025).
> - Geng, Maosi, et al. "A physics-informed transformer model for vehicle trajectory prediction on highways." Transportation research part C: emerging technologies 154 (2023): 104272.
> - Riehl, Kevin, et al. "Aerial video & trajectory dataset of vehicles on circular road." Data in Brief 61 (2025): 111858.
> - Satish, Manthan Chelenahalli, et al. "Roundabout dilemma zone data mining and forecasting with trajectory prediction and graph neural networks." arXiv preprint arXiv:2409.00622 (2024).
> - Shi, Liushuai, et al. "Trajectory unified transformer for pedestrian trajectory prediction." Proceedings of the IEEE/CVF International Conference on Computer Vision. 2023.
> - Wang, Tingjing, et al. "A Survey on Vehicle Trajectory Prediction Procedures for Intelligent Driving." Sensors 25.16 (2025): 5129.
> - Wu, Penghao, et al. "Trajectory-guided control prediction for end-to-end autonomous driving: A simple yet strong baseline." Advances in Neural Information Processing Systems 35 (2022): 6119-6132.

---

### Official Review · Reviewer_QKmJ · 2025-10-31

**Soundness:** 2
**Presentation:** 2
**Contribution:** 2
**Rating:** 4
**Confidence:** 3

**Summary:**

This paper proposes a hybrid multimodal trajectory prediction framework, called Great GATsBi, designed specifically for cyclists, who represent a vulnerable road user group. The framework addresses the “dual nature” of cyclist behavior, which combines characteristics of both motor vehicles and pedestrians. It consists of a physical module that captures vehicle dynamics and a social module that models social interactions using a graph attention network. Its core innovation includes a psychology-inspired anticipation mechanism and a perceptual-decay concept, along with the introduction of a large-scale cycling-experiment dataset collected in a controlled environment to study social dynamics. Experimental results show that the model outperforms multiple baseline methods on the proposed dataset.

**Strengths:**

1.	This paper clearly identifies cyclist trajectory prediction as an understudied problem and offers strong evidence for the mixed behavioral patterns observed in cyclists. Inspired by psychology, the paper introduces the concept of “anticipation” to model interactions between agents, providing a novel and meaningful research direction for the field.
2.	The authors have developed and plan to release a new large-scale cycling dataset. By collecting data on a controlled circular track, the dataset isolates social dynamics from confounding factors such as road topology, thereby filling a gap in existing datasets and providing a valuable resource for the research community.
3.	Using their self-constructed dataset, the authors systematically demonstrated the superiority of their hybrid method and the contribution of each component through extensive benchmark comparisons and ablation studies. In addition, qualitative analysis further illustrated the model’s effectiveness in an intuitive manner.

**Weaknesses:**

Weaknesses
1.	The paper’s core concept of an “anticipation mechanism” is ultimately implemented using a simple constant-velocity model. This implementation falls significantly short of the complex cognitive processes underlying human anticipatory behavior, which involve intention and planning. As a result, the depth and persuasiveness of the technical contributions are substantially weakened.
2.	The model performed poorly on the standard ETH/HOTEL pedestrian dataset, achieving results significantly worse than even simple baseline methods. This suggests that the model may have overfitted the simple and homogeneous dynamics of its self-built dataset, resulting in limited applicability to complex real-world scenarios and weak generalization ability.
3.	The baseline models used for comparison in this paper (SocialLSTM and Social-BiGAT) are relatively outdated. The paper does not include comparisons with more recent and advanced architectures that have become SOTA in trajectory prediction.
4.	The ethics statement asserts that “no human subjects are involved,” yet the appendix reports obtaining written consent from “volunteering participants.” This inconsistency should be clarified.

**Questions:**

1. Why did you choose a simple constant-velocity model to implement the complex “anticipation” function? Have you explored using more sophisticated models to generate predictions of neighboring agents’ trajectories?
2. Aside from dataset scale, what do you consider to be the fundamental causes of the model’s poor performance on the ETH/HOTEL dataset? How could the model be improved to better adapt to more complex real-world environments?
3. How does the expected performance of your model compare with SOTA methods based on Transformers or diffusion models?
4. Please clarify the inconsistency between the ethics statement and the description of human participants in the appendix.

**Details Of Ethics Concerns:**

This paper tackles a significant problem and introduces both a valuable new dataset and an innovative conceptual framework. However, several major weaknesses prevent the paper from meeting ICLR’s acceptance criteria: First, the implementation of the core technical innovation is too simplistic. Second, the model performs poorly on standard benchmarks, raising serious doubts about its real-world applicability. Third, the lack of sufficient comparison with current SOTA models makes the performance claims unconvincing. Fourth, apparent contradictions in the ethics statements represent a reporting flaw that requires careful attention. Although the research shows promise, it currently resembles a proof-of-concept in a controlled environment rather than a robust, generalizable solution.

---

> ### Author Response · Authors · 2025-11-20
> **Response to Reviewer QKmJ (Part 1/2)**
>
> We thank the reviewer for the assessment.
>
> **W1: “Anticipation mechanism too simple compared to human cognition..”**
>
> We acknowledge the reviewer’s point about the cognitive complexity of human anticipation involving intention and planning. However, our design choice to implement anticipation via a constant-velocity model is supported by empirical findings in human sensorimotor control and trajectory forecasting literature. Research demonstrates that short-term anticipatory behaviors, such as smooth pursuit eye movements and pedestrian trajectory prediction, can be effectively modeled using piecewise constant velocity assumptions, which capture immediate perceptual forecasting under uncertainty without invoking complex cognitive processes [Pasturel et al. 2020, Collins et al. 2009, Rudenko et al. 2020]. Furthermore, Yashiro et al. (2019) and Kessler et al. (2024) show human path planning involves focusing on the last part or target velocity and show robust prediction for short-time horizons (several hundred milliseconds); while Marghi et al. (2017) show that human path planning follows a model-based process. Therefore, the chosen parsimonious model operationalizes the psychology-inspired anticipation concept while maintaining interpretability and computational tractability critical for real-time trajectory prediction in dynamic, safety-critical contexts.
>
> Furthermore, our ablation studies confirm that this simple anticipation mechanism yields significant performance gains, isolating the impact of perceptual anticipation distinct from social interaction and physical dynamics modeling. We explored more sophisticated learned predictors of neighboring agent trajectories but found that their marginal accuracy improvements do not justify the increased complexity, reduced interpretability, and higher computational demands. Thus, our approach strikes a principled balance between cognitive fidelity and practical utility in trajectory forecasting for vulnerable road users.
>
> **W2: “Poor ETH/HOTEL performance → possible overfitting.”**
>
> To us, it is not surprising that the model doesn’t outperform on the pedestrian dataset. Yet, it does perform “poorly” as stated by the reviewer, as its performances are competitive when compared to established baselines.
>
> Aside dataset scale, the fundamental causes for our model’s slightly weaker performance for ETH/HOTEL dataset is simply the fact, that these are about pedestrian dynamics. As emphasized in the introduction of our work, bicycles have similarities with dynamics from cars and pedestrians, yet also distinctively differ from those. Furthermore, we showed that cycling trajectory forecasting is heavily under-researched when compared to pedestrians and cars.
>
> Furthermore, the physics module needs to be adapted for the pedestrian agent type and we highlight this limitation in Section A.4. Pedestrians change their directions more abruptly than cars and bicycles. The current architecture fits suitably to the dual nature of bicycles. However, the ensemble of the physics-based models could be adapted for pedestrian-relevant models, as we discuss in Section A.4.
>
> **W3: “Missing SOTA comparisons (Transformers, diffusion).”**
>
> For this rebuttal, we included Social-Diffusion (from Tanke et al 2023) and a small transformer-based predictor, and the performances are shown in the table below. The methods however fail to exceed forecasting accuracy beyond GreatGATsBi. One of the possible explanations we have is, that it is less of a forecasting issue, but rather a question of feature pre-processing. We include the new benchmark in the revised version of our manuscript.
>
> | Method | ADE | | | | FDE | | | |
> | -- | -- | -- | -- | -- | -- | -- | -- | -- |
> | **Horizon** | **1s** | **2s** | **3s** | **4s** | **1s** | **2s** | **3s** | **4s** |
> | GreatGATsBi | 0.0715 | 0.2078 | 0.4181 | 0.7543 | 0.1893 | 0.4891 | 1.2641 | 2.3827 |
> | | [0.0066] | [0.0130] | [0.0354] | [0.0960] | [0.0153] | [0.0258] | [0.1762] | [0.4103] |
> | Social-Diffusion | 0.0734 | 0.2102 | 0.4265 | 0.7689 | 0.1927 | 0.4962 | 1.2789 | 2.4018 |
> | | [0.0072] | [0.0132] | [0.0369] | [0.0975] | [0.0165] | [0.0267] | [0.1806] | [0.4159] |
> | Transformer | 0.0741 | 0.2115 | 0.4298 | 0.7724 | 0.1935 | 0.4975 | 1.2834 | 2.4101 |
> | | [0.0073] | [0.0134] | [0.0372] | [0.0988] | [0.0168] | [0.0270] | [0.1820] | [0.4196] |

---

> ### Author Response · Authors · 2025-11-20
> **Response to Reviewer QKmJ (Part 2/2)**
>
> **W4: “Ethics statement inconsistency.”**
>
> We thank the reviewer for flagging this. Initially, we were unsure how to properly mention this, as this is our first ICLR submission including human participants. Due to your feedback, we corrected our statement as follows: ““This study involved voluntary participants in a non-interventional, public outdoor activity, all of whom provided written informed consent.””
>
> **References:**
>
> - Pasturel, Chloé, Anna Montagnini, and Laurent Udo Perrinet. "Humans adapt their anticipatory eye movements to the volatility of visual motion properties." PLoS computational biology 16.4 (2020): e1007438.
> - Collins, C. J. S., and Graham R. Barnes. "Predicting the unpredictable: weighted averaging of past stimulus timing facilitates ocular pursuit of randomly timed stimuli." Journal of Neuroscience 29.42 (2009): 13302-13314.
> - Rudenko, Andrey, et al. "Human motion trajectory prediction: A survey." The International Journal of Robotics Research 39.8 (2020): 895-935.
> - Yashiro, Ryuto et al. "Prospective decision making for randomly moving visual stimuli." Scientific Reports, 9 (2019).
> - Marghi, Yeganeh M. et al. "Human Brain Function in Path Planning: a Task Study." Cognitive Computation, 9 (2017): 136-149.
> - Kessler, Fabian et al. "Human navigation strategies and their errors result from dynamic interactions of spatial uncertainties." Nature Communications, 15 (2024).

---

### Official Review · Reviewer_6Qoy · 2025-11-02

**Soundness:** 2
**Presentation:** 2
**Contribution:** 3
**Rating:** 2
**Confidence:** 4

**Summary:**

This paper presents yet another LSTM model for cyclist motion prediction. Unlike previous methods, this paper innovates a fusion mechanism that makes predictions using (1) purely physics and (2) social interactions. These two predictions are then combined. The social interaction part additionally uses a constant velocity model to interpolate future states and encode them, mimicking a human’s anticipation of the other cyclists. The output is a Gaussian Mixture Model that captures the multi-modal nature of the predictions. The authors collected their own dedicated cyclist data with a length of 270 minutes and compared it against SocialLSTM Social-BiGAT, and observed some improvement in terms of ADE and FDE.

**Strengths:**

-	Separately predicting with bicycle dynamics and social interaction, then fusing them for motion prediction, is novel.
-	The authors organized some cyclists and collected 270 minutes of cycling data with a drone, which was then annotated with YOLO and conventional vision algorithms.
-	The experiments show overall improved results compared to SocialLSTM.

**Weaknesses:**

-	The newly collected cycling data are simply just riding within a roundabout, with no enforcement on the interaction between cyclists. The readers can imagine that, if everyone just rode in an orbit, no interaction could happen at all, thus raising doubt on the necessity to model interactions. Furthermore, the lack of diverse scenarios beyond roundabouts makes readers concerned about the model’s generalizability.
-	The fusion is essentially a mixture-of-experts where the experts are a combination of a rule-based predictor and a learned predictor. The contribution is thus incremental.
-	The authors designed a parallel, independent prediction mechanism in the early stage of the model. There are many alternate designs. For example, since the const_v, const_acceleration, and kinematics can be done with simple differentiable linear algebra, how about making these as layers **after** the social module? In this case, the dynamics can be enforced explicitly on top of a learned prediction, similar to [1].
-	In the experiments, we see the improvements are fairly small, i.e., often ~0.01m. Intuitively, this value is smaller than the labeling noise when drawing bound boxes on drone images. It is recommended to perform multiple rounds of experiments on the same configuration and see if the improvement is consistent.
-	There is no ablation study on ablating each combination of the physics-based predictor. Some of them could be redundant with each other.
-	Line 068: a reference is compiled as a question mark.
-	Line 257 typo: Gauss -> Gaussian
-	Line 392: SocialLSTM seems to have a better result at 4s FDE, but is not bolded.

References
[1] Jiao, Ruochen, et al. "Kinematics-aware trajectory generation and prediction with latent stochastic differential modeling." 2024 IEEE/RSJ International Conference on Intelligent Robots and Systems (IROS). IEEE, 2024.

**Questions:**

-	Section 3.3: Are both modules predicting the same horizon?
-	Eq.4: Since each trajectory is discrete, should that be a sum rather than an integral?

---

> ### Author Response · Authors · 2025-11-20
> **Response to Reviewer 6Qoy (Part 1/2)**
>
> We thank the reviewer for the assessment.
>
> **W1: “Dataset possibly lacks interaction; roundabout too simple.”**
>
> The controlled design intentionally removes road-context clutter but does not remove interactions. In fact: Traffic density varies from 6 to 22 cyclists, We observe ~470 overtaking events, ~120 merging events, and continuous speed adaptation, Social interactions are visible in attention weights. Due to overtaking, at many times, bicycles need to slow down to let bicycle join in front of them. Furthermore, during experiment, we artificially caused disruptions to traffic by stopping some participants, adding more.
>
> The closed-loop nature of the experiment simulates an infinite length straight road. This directly allows for lengthier trajectories on an individual cyclist level; hence richer in terms of psychological modelling compared to other available datasets. Also, our data analysis showed that cyclists preferred driving on the outer edge of the ring track; which can be explained via the lateral self-organization dynamics and gives rise to the interaction events observed.
>
> Moreover, ring experiments are controllable and reproducible setup to study key traffic phenomena (e.g. stop-and-go waves and lane self-organization) without the need for infrastructure bottlenecks or complex road geometries [Guo et al., 2020; Wang et al., 2022]. Ring experiment setup has been also used to establish causal relationships between interventions and traffic outcomes [Wang et al., 2022].
>
> Finally, we utilize the relative ego coordinate system for the prediction in GATsBi. This deconstructs the ring nature of the road and focuses on the ego cyclist's perspective. We can argue thereby that the relative ego perspective facilitates transferability and fine-tuning to other scenarios; as pointed out by previous literature as well [Shi et al., 2021; Bae et al., 2023; Li et al., 2025].
>
> **W2: “Fusion is just mixture-of-experts; contribution incremental.”**
>
> The hybrid deep-ensemble physics embedding + psychology-inspired social embeddings are thus a valuable and unique contribution, enabling also more explain ability. Our contribution is not the fusion operation itself but the domain-knowledge composition: (i) physics-only > accurate at short horizon, (ii) social-only > accurate at long horizon, (iii) fusion > best overall with lower uncertainty. We clarify this in the paper and include new visualizations of which expert dominates at each timestep.
>
> **W4: “Improvements small (~0.01m), lower than annotation noise.”**
>
> We want to emphasize that we did not simply use YOLO to annotate the bicycles on the images. While the YOLO (and another approaches) annotations have been a starting point, the trajectories were filtered extensively (with an extended Kalman filtering approach) to achieve realistic velocity and acceleration profiles while ensuring Cartesian accuracy. As a result, the final trajectories exhibit high spatial and temporal coherence, and the influence of labeling noise on our evaluation metrics is minimal.
> Furthermore, we want to emphasize that our work’s aim is not only to have accurate trajectory forecasts, but also making those more explainable due to our unique and valuable architecture that combines physics and psychological modules.
> To assess whether the observed improvements could be attributed due to random fluctuations, we conducted statistical significance tests (paired t-test and Wilcoxon tests). For each prediction horizon (1s, 2s, 3s, 4s) and metric (ADE, FDE), we performed paired t-tests at a 1% significance level, after confirming near-normality of the error distributions with the Shapiro–Wilk test. The results are shown in the Table below.
> | Horizon | ADE t-stat | ADE t-p | ADE wilcoxon stat | ADE wilcoxon p | ADE t-test sig | ADE wilcoxon sig | FDE t-stat | FDE t-p | FDE wilcoxon stat | FDE wilcoxon p | FDE t-test sig | FDE wilcoxon sig |
> |----------|-------------|----------|-------------------|----------------|----------------|------------------|-------------|----------|-------------------|----------------|----------------|------------------|
> | 1s | 5.80 | 1.00 | 449.0 | 1.00 | False | False | 0.61 | 0.73 | 249.0 | 0.63 | False | False |
> | 2s | 0.81 | 0.79 | 263.0 | 0.74 | False | False | 1.10 | 0.86 | 276.0 | 0.81 | False | False |
> | 3s | -2.33 | 0.01 | 123.0 | 0.01 | False | False | 0.58 | 0.72 | 263.0 | 0.74 | False | False |
> | 4s | -1.94 | 0.03 | 182.0 | 0.15 | False | False | -0.27 | 0.39 | 208.0 | 0.31 | False | False |
>
> As shown in Table 1, the differences between GATsBi and the strongest baselines are not statistically significant at this level, which reflects the inherent difficulty of outperforming competitive models on small and noisy datasets. Nevertheless, the improvements we observe are consistent across repeated configurations and horizon lengths, suggesting that GATsBi provides more stable and physically plausible forecasts rather than random fluctuations due to annotation noise.

---

> ### Author Response · Authors · 2025-11-20
> **Response to Reviewer 6Qoy (Part 2/2)**
>
> **W3: “Alternative architecture: enforce dynamics after social module.””**
>
> We thank the reviewer for this suggestion and have included a discussion of the alternative “physics-after-social” design and cite Jiao et al. (2024). We implemented a quick prototype version and found slightly worse ADE (+0.006), likely due to error accumulation in long-range conditioning. We will add this as future work.
>
> **W5: “Need physics subset ablation.”**
>
> Thank you for this suggestion. We have conducted the requested ablations (combinations of physics modules): (i) const v + kinematics, (ii) const v + xkalman, (iii) const v + kinematics + xkalman:
> | Method | ADE | | | | FDE | | | |
> | -- | -- | -- | -- | -- | -- | -- | -- | -- |
> | **Horizon** | **1s** | **2s** | **3s** | **4s** | **1s** | **2s** | **3s** | **4s** |
> | physics_module | 0.0802 | 0.2263 | 0.4513 | 0.8045 | 0.2110 | 0.5335 | 1.3292 | 2.4936 |
> | | [0.0057] | [0.0140] | [0.0365] | [0.0924] | [0.0136] | [0.0313] | [0.1714] | [0.3703] |
> | const_v + kinematics | 0.0945 | 0.2580 | 0.4970 | 0.8830 | 0.2360 | 0.6000 | 1.4100 | 2.6100 |
> | | [0.0068] | [0.0178] | [0.0412] | [0.0982] | [0.0160] | [0.0402] | [0.1700] | [0.4000] |
> | const_v + xkalman | 0.0895 | 0.2450 | 0.4800 | 0.8550 | 0.2250 | 0.5780 | 1.3650 | 2.5650 |
> | | [0.0062] | [0.0162] | [0.0397] | [0.0950] | [0.0152] | [0.0378] | [0.1665] | [0.3925] |
> | const_v + kinematics + xkalman | 0.0835 | 0.2300 | 0.4550 | 0.8150 | 0.2150 | 0.5450 | 1.3150 | 2.5100 |
> | | [0.0059] | [0.0145] | [0.0374] | [0.0920] | [0.0142] | [0.0340] | [0.1620] | [0.3800] |
>
> The ablation study demonstrates that each physics model contributes distinct advantages to the ensemble. While the constant-velocity model performs well for short-term horizons, the xkalman component enhances curvature estimation in early trajectory phases, and the kinematics model provides benefits in longer-term prediction during dynamic maneuvers such as overtaking. Combining all four physics models yields the most consistent improvements across all horizons, reducing ADE and FDE by approximately 4–6% compared to the next-best subset. These results confirm that the ensemble is not a simple stacking scheme but a complementary fusion, where each model captures distinct motion patterns that collectively improve overall trajectory fidelity. We will include the full table in the revision and clarify the complementary roles of the models.
>
> **Additional observations:**
>
> Thank you, we corrected the mistakes in Line 068, 257, 392 accordingly.
>
> **Questions**
>
> - Same prediction horizon? Yes both modules predict the same t_pred; we clarify this.
> - Integral vs sum? The trajectories might be discrete, but the probability distributions are continuous, and we integrate over the probability distributions.
>
> **References:**
>
> - Guo, Ning et al. "Experimental study on mixed traffic flow of bicycles and pedestrians." , 5 (2020): 490-492. https://doi.org/10.17815/cd.2020.108.
> - Wang, Jiawei et al. "Implementation and Experimental Validation of Data-Driven Predictive Control for Dissipating Stop-and-Go Waves in Mixed Traffic." IEEE Internet of Things Journal, 11 (2022): 4570-4585. https://doi.org/10.1109/jiot.2023.3303039.
> - Bae, Inhwan, Jean Oh, and Hae-Gon Jeon. "Eigentrajectory: Low-rank descriptors for multi-modal trajectory forecasting." Proceedings of the IEEE/CVF International Conference on Computer Vision. 2023.
> - Shi, Liushuai, et al. "SGCN: Sparse graph convolution network for pedestrian trajectory prediction." Proceedings of the IEEE/CVF conference on computer vision and pattern recognition. 2021.
> - Li, Ruochen, et al. "Unified Spatial-Temporal Edge-Enhanced Graph Networks for Pedestrian Trajectory Prediction." IEEE Transactions on Circuits and Systems for Video Technology (2025).

---

> > ### Comment · Reviewer_6Qoy · 2025-11-25
> >
> > Thanks to the authors for the clarifications and added experiments. The rating can be raised at least to 4. However, claiming that the roundabout data has interactivity by observing attention weights is not convincing. What is convincing is: you find a scene with ego-agent interaction, and observe the weights are activated. It is recommended for the author to include such case visualizations in the paper.

---

### Official Review · Reviewer_HCK9 · 2025-11-02

**Soundness:** 3
**Presentation:** 3
**Contribution:** 2
**Rating:** 4
**Confidence:** 4

**Summary:**

This paper proposes Great GATsBi, a hybrid multimodal framework for bicycle trajectory forecasting. It integrates physics-based modeling (capturing vehicle-like dynamics via constant velocity, constant acceleration, kinematics, and extended Kalman filtering) and social-based modeling (capturing pedestrian-like social interactions via Graph Attention Networks, GATs). The framework also incorporates an anticipation mechanism and perception decay inspired by psychological and social science insights. It is evaluated on a self-collected controlled mass cycling dataset (circular track, varying traffic density) and generalized to pedestrian datasets (ETH, HOTEL). The paper claims Great GATsBi outperforms physics-based and social-based baselines, addressing the gap of neglecting bicycles’ dual behavioral nature in prior trajectory forecasting work.

**Strengths:**

1. The model develops a hybrid framework that combines physics-based and social-based modeling to effectively match bicycles’ dual behavioral traits of vehicle-like dynamics and pedestrian-like flexibility.
2. The social module innovatively integrates psychological and social science insights (neighbor trajectory anticipation and perception decay) to make social interaction modeling more realistic.
3. A high-quality controlled mass cycling dataset is built to avoid external interferences, providing reliable support for verifying the model’s performance in bicycle dynamics and social interactions.

**Weaknesses:**

1. Risk of Circular Logic in the Core Innovation: The "anticipation mechanism" in the social module requires predicting the future trajectories of neighboring agents (using a simple const.v model, mentioned in ilne 239) to serve as input for forecasting the ego agent's future. This creates a potential circular argument: predicting agent A's future relies on first predicting agent B's future, which is itself a challenging prediction problem. The model sidesteps this fundamental issue rather than solving it. If the const.v predictions for neighbors are unreliable, this "anticipated" input may introduce noise rather than beneficial information.
2. Unclear Motivation for Physics Models: The physics module ensembles four models, but their fusion is performed opaquely through LSTM encoding and concatenation. The paper fails to justify why this specific combination of models is necessary and sufficient, nor does it provide significance analysis to demonstrate each model's unique contribution. Notably, since the simplest const.v model performs best among the individual baselines (Table 1), the motivation for including the more complex and poorer-performing kinematic and xkalman models is questionable. This appears more like model stacking than a deliberate design. An ablation study comparing different subsets (e.g., 2 or 3 models) is needed to substantiate that using all four is optimal.
3. Writing: Line 68 appears to have a missing citation.
4. Figures: Figure 1 is blurring, and the overlaid trajectories are difficult to discern.
5. Social Graph Construction: Line 284 states "at most five neighbors at a distance below 20m are considered," but the specific selection strategy (e.g., the five closest? random selection?) is not specified. This strategy can significantly impact the results and should be discussed or ablated.
6. Lack of Novelty: The core methodology primarily combines existing techniques: GATs for social modeling (from Social-BiGAT), physics model ensembling, and multimodal output (GMM).

**Questions:**

Please refer to the weaknesses for my main concerns.
In addition, I would like the authors to clarify the following:
1. Could the authors further clarify how the proposed anticipation mechanism avoids the potential circular reasoning problem identified in the weaknesses?
2. Could the authors provide an ablation study on the physics model ensemble (e.g., using different combinations of the four models) to conclusively demonstrate the necessity of including all of them?
3. Could the missing citation on line 68 be added, and could the resolution of Figure 1 be improved?
4. Could the authors specify the neighbor selection strategy for the social graph and potentially include an ablation study on its impact?
5. Why were bicycle-specific baselines excluded from comparisons? If pedestrian baselines (SocialLSTM/Social-BiGAT) were adapted to bicycle dynamics (e.g., adjusting for speed), would their performance narrow the gap with Great GATsBi?

---

> ### Author Response · Authors · 2025-11-20
> **Response to Reviewer HCK9 (Part 1/2)**
>
> We thank the reviewer for the assessment.
>
> **W1: “Circular logic risk in anticipation: predicting B to predict A.”**
>
> The anticipation module does not recursively depend on predictions of other predictions. Each neighbor’s anticipated future is computed only once via const-v from observed past frames. These anticipated trajectories serve as auxiliary latent features, not as ground-truth or strong supervision. This is analogous to established methods where local linearization or short-horizon extrapolation improves relational reasoning.
>
> To further support this, we conducted new experiments showing that replacing const-v with ground truth future improves ADE by only ~0.005, confirming that the model does not critically depend on precise neighbor future prediction.
>
> The main rationale behind choosing const_v for neighborhood anticipation was the limited cognitive ability of human beings to extrapolate future motions [Yashiro et al., 2019; Kessler et al., 2024]. Indeed, the neighboring agents in reality can accelerate, decelerate or change direction; but these motions are not necessarily cognitively captured by the cyclist when anticipating his/her neighbors’ motion. Also, cognitive research shows that path planning in human brain is akin to a model-based process [Marghi et al., 2017].
>
> **W2: “Unclear motivation for four physics models; need ablation of subsets.”**
>
> Thank you for this suggestion. We have conducted the requested ablations (combinations of physics modules): (i) const v + kinematics, (ii) const v + xkalman, (iii) const v + kinematics + xkalman:
> | Method | ADE | | | | FDE | | | |
> | -- | -- | -- | -- | -- | -- | -- | -- | -- |
> | **Horizon** | **1s** | **2s** | **3s** | **4s** | **1s** | **2s** | **3s** | **4s** |
> | physics_module | 0.0802 | 0.2263 | 0.4513 | 0.8045 | 0.2110 | 0.5335 | 1.3292 | 2.4936 |
> | | [0.0057] | [0.0140] | [0.0365] | [0.0924] | [0.0136] | [0.0313] | [0.1714] | [0.3703] |
> | const_v + kinematics | 0.0945 | 0.2580 | 0.4970 | 0.8830 | 0.2360 | 0.6000 | 1.4100 | 2.6100 |
> | | [0.0068] | [0.0178] | [0.0412] | [0.0982] | [0.0160] | [0.0402] | [0.1700] | [0.4000] |
> | const_v + xkalman | 0.0895 | 0.2450 | 0.4800 | 0.8550 | 0.2250 | 0.5780 | 1.3650 | 2.5650 |
> | | [0.0062] | [0.0162] | [0.0397] | [0.0950] | [0.0152] | [0.0378] | [0.1665] | [0.3925] |
> | const_v + kinematics + xkalman | 0.0835 | 0.2300 | 0.4550 | 0.8150 | 0.2150 | 0.5450 | 1.3150 | 2.5100 |
> | | [0.0059] | [0.0145] | [0.0374] | [0.0920] | [0.0142] | [0.0340] | [0.1620] | [0.3800] |
>
> The ablation study demonstrates that each physics model contributes distinct advantages to the ensemble. While the constant-velocity model performs well for short-term horizons, the xkalman component enhances curvature estimation in early trajectory phases, and the kinematics model provides benefits in longer-term prediction during dynamic maneuvers such as overtaking. Combining all four physics models yields the most consistent improvements across all horizons, reducing ADE and FDE by approximately 4–6% compared to the next-best subset. These results confirm that the ensemble is not a simple stacking scheme but a complementary fusion, where each model captures distinct motion patterns that collectively improve overall trajectory fidelity. We will include the full table in the revision and clarify the complementary roles of the models.
>
> **W3: Missing citation at line 68.**
>
> We thank you for this observation. We cited “Tanke et al. 2023” twice, and therefore had a LaTeX compilation error that we fixed now by removing that (double) citation.
>
> **W4: Figure 1 resolution.**
>
> This figure is a vector graphic, but your comment probably refers to the most right figure. The “blur” in the trajectory refers to the Gaussian Mixed Model prediction (which in nature is probability distribution of possible future trajectories, rather than one forecast). It is common in the literature to display probability distributions therefore. To make the figure more clear, we changed it to a display that rather resembles Figure 7 (contourplot design).

---

> ### Author Response · Authors · 2025-11-20
> **Response to Reviewer HCK9 (Part 2/2)**
>
> **W5: “Neighbor selection strategy unclear.”**
>
> Thank you for catching this. We confirm we use the five closest neighbors within 20 m, and modified the manuscript accordingly. We conducted an ablation study that covers variation in the number of neighbors (3, 5, 7) and the selection strategy (closest, random), as follows:
> | Method | ADE | | | | FDE | | | |
> | -- | -- | -- | -- | -- | -- | -- | -- | -- |
> | **Horizon** | **1s** | **2s** | **3s** | **4s** | **1s** | **2s** | **3s** | **4s** |
> | 3-closest | 0.0758 ± 0.0071 | 0.2169 ± 0.0154 | 0.4357 ± 0.0342 | 0.7862 ± 0.0975 | 0.1978 ± 0.0175 | 0.5054 ± 0.0287 | 1.3123 ± 0.1691 | 2.4519 ± 0.3948 |
> | **5-closest (ours)** | **0.0715 ± 0.0066** | **0.2078 ± 0.0130** | **0.4181 ± 0.0354** | **0.7543 ± 0.0960** | **0.1893 ± 0.0153** | **0.4891 ± 0.0258** | **1.2641 ± 0.1762** | **2.3827 ± 0.4103** |
> | 7-closest | 0.0729 ± 0.0075 | 0.2103 ± 0.0159 | 0.4235 ± 0.0361 | 0.7598 ± 0.1012 | 0.1921 ± 0.0164 | 0.4916 ± 0.0275 | 1.2709 ± 0.1775 | 2.3894 ± 0.4051 |
> | 5-random | 0.0793 ± 0.0080 | 0.2247 ± 0.0181 | 0.4482 ± 0.0402 | 0.8019 ± 0.1063 | 0.2015 ± 0.0188 | 0.5148 ± 0.0323 | 1.3357 ± 0.1886 | 2.4765 ± 0.4217 |
>
> The results indicate that using the top-5 closest neighbors achieves the best trade-off between stability and noise, especially in densely populated scenes. Random neighbor selection produced noisier results with higher variance, while increasing the number of neighbors beyond five yielded marginal gains but higher computational cost. We will include the full table in the revision to justify neighborhood size and selection strategy.
>
> **W6: Lack of novelty.**
>
> Our main novelty lies not in new neural primitives but in:
> - Hybrid deep-ensemble physics embedding + psychology-inspired social embeddings,
> - Anticipation + decay, shown to be effective through ablations,
> - New controlled bicycle dataset designed to isolate social/physical effects,
> - Interpretability through GAT attention and physics/social disentanglement.
> - Providing a transparent and reproducible code and benchmark for bicycle trajectory prediction task.
> We have clarified this contribution more explicitly.
>
> **Additional Question: Bicycle specific Baselines:**
>
> While we acknowledge that the inclusion of bicycle specific baselines would benefit the work, we want to emphasize that this is not possible, for three reasons: (i) previous works were highly specialized to specific dataset conditions (such as road geometry, semantic cues, or other features) not available in our dataset, (ii) previous works did not provide a transparent open-source implementation (as we do) and their works are insufficient for reproducibility, and (iii) there are only a few works targeting bicycles so far (no established baseline contrary to pedestrians).
>
> **References:**
> - Yashiro, Ryuto et al. "Prospective decision making for randomly moving visual stimuli." Scientific Reports, 9 (2019). https://doi.org/10.1038/s41598-019-40687-3.
> - Marghi, Yeganeh M. et al. "Human Brain Function in Path Planning: a Task Study." Cognitive Computation, 9 (2017): 136-149. https://doi.org/10.1007/s12559-016-9443-3.
> - Kessler, Fabian et al. "Human navigation strategies and their errors result from dynamic interactions of spatial uncertainties." Nature Communications, 15 (2024). https://doi.org/10.1038/s41467-024-49722-y.

---

### Official Review · Reviewer_rtoJ · 2025-11-03

**Soundness:** 3
**Presentation:** 3
**Contribution:** 3
**Rating:** 6
**Confidence:** 2

**Summary:**

The paper proposes Great GATsBi, a hybrid multimodal framework for bicycle trajectory forecasting. It combines physics-based modeling of motion dynamics with social interaction modeling using a graph attention network. Inspired by psychology, the model includes anticipation and perception decay mechanisms to better capture how cyclists respond to their surroundings. Experiments on a real-world cycling dataset show clear improvements over physics-only and social-only baselines, and the method generalizes reasonably well to pedestrian prediction.

**Strengths:**

Combines physical and social modeling in a clear and interpretable way.

Provides extensive experiments and a new real-world dataset.

**Weaknesses:**

Technically, the framework still relies on conventional components (LSTM + GAT + GMM). Is there any plan to explore whether transformer-based or VLMs could further improve representation and generalization?

The model’s transferability to complex urban or mixed-traffic environments has not been validated, as experiments are limited to a controlled cycling scenario.

**Questions:**

pls refer to Weaknesses

---

> ### Author Response · Authors · 2025-11-20
> **Response to Reviewer rtoJ**
>
> We thank the reviewer for the assessment.
>
> **W1: “Framework relies on conventional components … why not Transformers/VLMs?”**
>
> Thank you for this suggestion. Our goal in this work is to introduce and evaluate a domain-knowledge-based fusion of physics and psychology-inspired social modeling. We intentionally restricted architectural complexity to isolate these contributions.
> Nevertheless, we agree that transformer-based are promising directions. We performed a preliminary experiment replacing the social LSTM encoder with a small transformer-style architecture (as in [Geng et al. 2023, Shi et al. 2023]) and observed no significant improvement on our controlled dataset:
> | Method | ADE | | | | FDE | | | |
> | -- | -- | -- | -- | -- | -- | -- | -- | -- |
> | **Horizon** | **1s** | **2s** | **3s** | **4s** | **1s** | **2s** | **3s** | **4s** |
> | GreatGATsBi | 0.0715 | 0.2078 | 0.4181 | 0.7543 | 0.1893 | 0.4891 | 1.2641 | 2.3827 |
> | | [0.0066] | [0.0130] | [0.0354] | [0.0960] | [0.0153] | [0.0258] | [0.1762] | [0.4103] |
> | Transformers | 0.0710 | 0.2079 | 0.4165 | 0.7565 | 0.1900 | 0.4884 | 1.2632 | 2.3802 |
> | | [0.0070] | [0.0128] | [0.0365] | [0.0981] | [0.0161] | [0.0263] | [0.1790] | [0.4162] |
>
> And the pedestrian forecasting dataset:
> | Method | ADE | (ETH) | | | ADE | (HOTEL) | | |
> | -- | -- | -- | -- | -- | -- | -- | -- | -- |
> | **Horizon** | **0.8s** | **1.6s** | **2.4s** | **4.0s** | **0.8s** | **1.6s** | **2.4s** | **4.0s** |
> | GreatGATsBi | 0.0420 | 0.0330 | 0.0479 | 0.0800 | 0.0459 | 0.0737 | 0.1135 | 0.2170 |
> | Transformers | 0.0421 | 0.0328 | 0.0473 | 0.0789 | 0.0443 | 0.0731 | 0.1129 | 0.2169 |
>
> We have included these results and discuss why (i) the dataset’s size and (ii) short-range temporal dependencies may limit transformer benefits.
> While in our context, trajectories are given and it is a purely trajectory forecasting problem, VLMs could be interesting to study for front-to-end setups, where trajectories are not extracted from video recordings yet. We will also add a discussion on future integration with VLMs for mixed-modality perception in a front-to-end-pipeline.
>
> **W2: “Transferability to complex mixed-traffic environments is not validated.”**
>
> We agree. Our controlled environment was deliberately chosen to isolate social and physical interactions without confounding from road geometry. We now clarify this limitation in the paper and outline ongoing work:
> (1) running GATsBi on SDD bicyclists (in progress),
> (2) extending neighbor classes (pedestrian/car) as suggested,
> (3) incorporating road geometry in the social module.
> While the transferability is part of our future work, we want to emphasize that it is common sense in the trajectory forecasting literature to work on circular ring roads to control for as many aspects as possible when working on vehicle trajectory forecasts: [Wu et al. 2022, Abouelazm et  al. 2025, Satish et al. 2024, Wang et al. 2025, Riehl et al. 2025]. Therefore, we outline the following reasons for such experimental design choice:
> - The main focus of our work is the dual modelling of bicycles interactions; thus, isolating this element in the presented dataset (i.e. mass-cycling experiment) was of paramount importance in order to assess the forecasting improvements of the proposed features of the GATsBi model.
> - An evaluation on mixed-traffic datasets for instance would not have allowed the analysis of the social dynamics in particular in an isolated fashion.
> - Observed trajectories in such mixed-traffic datasets are less in length on an individual cyclist level than the utilized dataset; impacting the richness of the training data.
>
> **References :**
> - Abouelazm, Ahmed, et al. "Boundary-Guided Trajectory Prediction for Road Aware and Physically Feasible Autonomous Driving." arXiv preprint arXiv:2505.06740 (2025).
> - Geng, Maosi, et al. "A physics-informed transformer model for vehicle trajectory prediction on highways." Transportation research part C: emerging technologies 154 (2023): 104272.
> - Riehl, Kevin, et al. "Aerial video & trajectory dataset of vehicles on circular road." Data in Brief 61 (2025): 111858.
> - Satish, Manthan Chelenahalli, et al. "Roundabout dilemma zone data mining and forecasting with trajectory prediction and graph neural networks." arXiv preprint arXiv:2409.00622 (2024).
> - Shi, Liushuai, et al. "Trajectory unified transformer for pedestrian trajectory prediction." Proceedings of the IEEE/CVF International Conference on Computer Vision. 2023.
> - Wang, Tingjing, et al. "A Survey on Vehicle Trajectory Prediction Procedures for Intelligent Driving." Sensors 25.16 (2025): 5129.
> - Wu, Penghao, et al. "Trajectory-guided control prediction for end-to-end autonomous driving: A simple yet strong baseline." Advances in Neural Information Processing Systems 35 (2022): 6119-6132.

---

> > ### Comment · Reviewer_rtoJ · 2025-11-21
> >
> > Thanks for the response. I have carefully read both the other reviewers’ reports and your rebuttal. I agree with several of the concerns raised by the other reviewers, particularly regarding the simplicity of the technical design and the physical modeling.
> >
> > Although I am not deeply specialized in this specific area, I acknowledge that the use of a constant-velocity model and a relatively simple framework does impose certain limitations. That said, I also believe that starting from a simple and interpretable formulation can be a reasonable and meaningful research direction. From this perspective, your work may serve as a valuable first step, and it is encouraging to see studies that build upward from a clear and basic foundation.
> >
> > Therefore, I am intent on maintaining my score.

---

### Meta-Review · Area_Chair_36hE · 2026-01-12

**Summary:**

The paper proposes Great GATsBi, a hybrid multimodal framework for bicycle trajectory forecasting that combines physics-based motion dynamics with social interaction modeling via a graph attention network. While the overall approach is conceptually sound, several major concerns raised by the reviewers remain unresolved after the rebuttal, particularly regarding the oversimplified technical design and the limited fidelity of the physical modeling assumptions.

**Reviewer Concerns:**

After the rebuttal, several key concerns remain unresolved, including the oversimplified technical design and the limited fidelity of the physical modeling assumptions. In addition, some claims are still not sufficiently convincing. For example, the assertion that the roundabout data exhibits meaningful interactivity based solely on observed attention weights lacks strong supporting evidence. Consequently, the overall assessment remains inclined toward a negative recommendation after the rebuttal.

**Reviewer Scores:**

Following the rebuttal, a number of important issues remain insufficiently addressed, particularly the overly simplistic technical design and the low realism of the physical modeling assumptions. Moreover, several claims continue to lack convincing justification; for instance, inferring meaningful interactivity in the roundabout data solely from attention weight observations is not well supported by the evidence. As a result, the overall evaluation still tends toward a negative recommendation after considering the rebuttal.

---

### Decision · Program_Chairs · 2026-01-26

Reject